behaviour/cognition/psychology

behavioural syndrome, innovation, cognitive syndrome, individual differences, problem solving, animal cognition

**Author for correspondence:**
Lisa P. Barrett
e-mail: lisapbarrett.lpb@gmail.com

# Links between personality traits and problem-solving performance in zebra finches (*Taeniopygia guttata*)

Lisa P. Barrett[1,2], Jessica L. Marsh[1], Neeltje J. Boogert[3], Christopher N. Templeton[4] and Sarah Benson-Amram[1,2,5,6]

[1]Department of Zoology and Physiology, University of Wyoming, Dept. 3166, 1000 E. University Ave, Laramie, WY 82071, USA
[2]Program in Ecology, University of Wyoming, Laramie, WY 82071, USA
[3]Department of Biosciences, University of Exeter, Penryn Campus, Penryn TR10 9FE, UK
[4]Department of Biology, Pacific University Oregon, 2043 College Way, Forest Grove, OR 97116, USA
[5]Department of Forest and Conservation Sciences, University of British Columbia, 3041-2424 Main Mall, Vancouver, British Columbia, Canada V6T 1Z4
[6]Department of Zoology, University of British Columbia, 4200-6270 University Blvd, Vancouver, British Columbia, Canada V6T 1Z4

LPB, 0000-0001-6072-3479; NJB, 0000-0002-1337-4365; CNT, 0000-0001-8701-3186; SB, 0000-0003-0147-7559

Consistent individual differences in behaviour across time or contexts (i.e. personality types) have been found in many species and have implications for fitness. Likewise, individual variation in cognitive abilities has been shown to impact fitness. Cognition and personality are complex, multidimensional traits. However, previous work has generally examined the connection between a single personality trait and a single cognitive ability, yielding equivocal results. Links between personality and cognitive ability suggest that behavioural traits coevolved and highlight their nuanced connections. Here we examined individuals' performance on multiple personality tests and repeated problem-solving tests (each measuring innovative performance). We assessed behavioural traits (dominance, boldness, activity, risk-taking, aggressiveness and obstinacy) in 41 captive zebra finches. Birds' scores for boldness and obstinacy were consistent over two years. We also examined whether personality correlated with problem-solving performance on repeated tests. Our results indicate that neophobia, dominance and obstinacy were related to successful solving, and less dominant, more obstinate birds solved the tasks quicker on average. Our results indicate the importance of examining multiple measures over a long period. Future work that identifies links between personality and innovation in non-model organisms may elucidate the coevolution of these two forms of individual differences.

# 1. Background

Recent research on animal personality explores how inter-individual consistency in behaviour (e.g. shy, bold, aggressive) across contexts (e.g. foraging, parental care and social interactions) may have fitness consequences for a variety of taxa, ranging from humans to birds to insects [1–4]. Animal personality has been linked to disease susceptibility, dispersal tendency, parental care, and other life-history outcomes [1]. Individual variation in animal cognition (i.e. how animals perceive, learn about and solve problems in their environment [5]) likely relates to personality types, forming a cognitive syndrome where, for instance, bolder, more aggressive, and more exploratory individuals gather information more quickly than shyer, less aggressive, less exploratory individuals [6]. Understanding the links between personality and cognition contributes to our understanding of the coevolution of behavioural responses; determining patterns between seemingly disparate traits can help predict how animals will respond to novel stimuli or changes to their environment [7]. Here we help elucidate how personality relates to innovative problem solving in animals.

Innovative problem solving is commonly measured in studies linking personality to behaviour (e.g. [8–11]; but see debate about whether innovation is an indicator of cognitive ability, e.g. [12,13]). Innovation, the ability to solve novel problems or to find new solutions to an old problem [14], is crucial for locating and using novel resources in a changing environment [15–17], because innovation involves exploiting the environment in a new way, such as shifting to foraging on a new resource [17]. For example, studies of foraging innovation in birds have often focused on measuring the ingestion of new food types or the use of new foraging techniques (e.g. [18]). To measure innovative tendencies of animals, researchers commonly present problem-solving tasks, such as novel foraging tasks that involve manipulating an apparatus to retrieve a food reward inside [7,19]. Personality traits correlate with problem-solving performance on these tasks across taxa, although the direction of the relationship varies [7,8,20,21]. Variation in personality is thought to be linked because of an underlying physiological response to change in environmental stimuli (i.e. stress response) [20], and this could explain how individuals approach and solve novel problems in their environment. 'Faster' (bolder, more active, and more aggressive) individuals may more quickly approach and solve a novel problem and be more likely to take risks compared to 'slower' (shyer, less active and less aggressive) individuals, for instance [2,8,21]. There has been a surge of interest into the amount of covariation between certain personality profiles and cognitive performance. A recent meta-analysis noted that the direction of the relationship between personality and cognitive traits remains unclear; this relationship is not only species- or sex-specific, but also depends on which personality and cognitive traits are measured and how they are assessed [22]. For example, less dominant male guppies were more likely to innovate a solution to a novel foraging task compared to more dominant males [23], which was also found in both male and female great tits [24]. These findings support the 'necessity drives innovation' hypothesis, where, due to metabolic requirements, less dominant individuals (who also receive fewer resources due to social competition) are forced to innovate novel foraging solutions [10,20,25,26]. On the other hand, *more* dominant chimpanzees spent more time interacting with a novel foraging task compared to their less dominant counterparts, and more dominant males were also more successful in solving the task [27]. Other studies have focused on the association between different personality traits, such as activity levels and neophobia (which have been shown to form a syndrome along a proactive–reactive axis [4]), and innovation, with mixed results (reviewed in [7]). Collecting multiple measures of personality would help to identify how individual differences in personality relate to individual differences in response to varied environmental contingencies [22]. A behavioural syndrome is a multidimensional collection of behavioural traits, and thus multiple personality traits should be investigated simultaneously to generate a deeper understanding of how personality relates to problem-solving performance [28]. For example, aggression is often related to boldness and activity, where more neophilic, aggressive and active individuals may solve a task faster because they sample the environment faster than more neophobic, less aggressive and less active individuals [2,6]. More work is needed to elucidate how multiple personality traits contribute to observed cognitive performance, including how a task is approached and solved. One of our primary objectives seeks to examine the relationship between multiple traits and innovativeness.

Moreover, we also require a better understanding of the *long-term* repeatability of behavioural measures, since most assessments of personality are conducted in short-term experiments, typically measuring only a few traits across a few days or weeks (but see, for example, [9,29,30] which measured select traits over longer periods of time). Individuals are more consistent over short

intervals than they are over longer intervals [31], so shorter-term experiments may inflate repeatability estimates and be biased towards certain life-history stages and environments (reviewed in [29,31]). Personality is defined by consistency of behaviour, but there are differences in consistency across time periods, so we require a greater understanding about the validity of personality traits across time [32]. Thus, another primary objective of this work was to examine this methodological gap—to measure a greater number of personality traits over a longer time frame in order to capture responses in diverse biological and environmental contexts.

Zebra finches (*Taeniopygia guttata*) are small, socially monogamous, passerine birds from Australia that live in large flocks and forage on grass seeds. Because they are easily kept in captivity, zebra finches have become a model system for studies of animal cognition, including song learning, and more recently for studies of animal personality (e.g. [27–29]), but none have looked at personality and innovation in conjunction. This missing link could be important for zebra finch fitness because food resources can be difficult to locate, and competitive ability could be important during interference competition over food resources in flocks [33]; moreover, innovation could be influenced by competitive ability or vice versa [10,34]. Individuals capable of accessing food resources (such as more dominant individuals), especially when food is ephemeral, may possess an advantage over other individuals to expand the amount or type of food available to them [25]. Indeed, previous work with zebra finches showed that more aggressive individuals (and not those in better body condition) were more dominant [33]. Alternatively, however, *less* dominant individuals could be more innovative, as predicted by the 'necessity drives innovation' hypothesis [10,20,25,26]. Thus, zebra finches are well-suited for tests of the 'necessity drives innovation' hypothesis. Also, results about the relative importance of personality traits for accessing novel resources (or accessing resources in novel ways) would inform our understanding of trade-offs between consistency in some traits (i.e. personality) and plasticity in cognition, which might apply to other group-living species that face novel challenges in their environment. We examined this link between personality and cognition (using three novel foraging tasks) in zebra finches. We also investigated temporal consistency of traits by examining the same personality traits across multiple time intervals. By examining these topics in a model organism that has already been the focus of several personality and cognition studies, we provide a foundation for future studies on other, understudied species.

We conducted tests of multiple behavioural traits, recording multiple measures of each trait, and repeated problem-solving tests that each measured innovative performance. We: (1) estimated both the short- and long-term repeatability of behavioural traits such as activity, neophobia, and obstinacy and (2) asked whether repeatable variation in these behavioural traits is associated with differences in problem-solving performance. To address question one, we predicted that these behavioural measures would be repeatable over both the short term and long term [33,35,36]. To address question two, we made a series of predictions based on what is currently known about zebra finch personality, and the link between personality and innovation more broadly (table 1). For example, activity has been measured with regard to predator–prey trade-offs, where active individuals might be prone to inappropriate activity in the presence of predators; i.e. higher activity results in higher feeding rates, but also higher predation risk [2]. In terms of cognition, more proactive individuals are faster at initial learning but slower to relearn when contingencies have changed. In other words, faster individuals more readily form routines but react slowly to changes in their environment, so making fast decisions might be beneficial over the short term as individuals gain more resources, but these quick decisions may be inaccurate or risky [43].

# 2. Methods

## 2.1. Subjects

Subjects were 41 unrelated, adult (same-age) zebra finches (21 males and 20 females) from Magnolia Bird Farm breeder (Anaheim, CA, USA). The birds were all acquired as young adults and started testing four months later. We identified individual birds by unique leg band colour and number combinations. Birds were initially kept in same-sex groups consisting of ten randomly chosen individuals. These groups were housed in flight cages (88.90 cm high × 81.30 cm wide × 53.34 cm deep) at the University of Wyoming. Eggs were removed to prevent reproduction. After a two-week habituation period to flight cage groups, and after completion of the Assessment of Dominance Hierarchies, birds were then housed in pairs with a same-sex individual in home cages (51.44 cm high × 99.06 cm wide × 50.17 cm deep) and

**Table 1.** Predicted associations between behavioural traits and innovation measured in this study.

| trait measured | predicted association with innovation | explanation |
|---|---|---|
| dominance | + or − | the effect could be in either direction. We predicted that more dominant birds would be better innovators because they would perform well at gaining access to food resources (*sensu* [32,35]), but see other research that has shown that this is not true in an asocial context [37]. Alternatively, subordinate individuals could be better innovators than dominant individuals, following the 'necessity drives innovation' hypothesis [23,24,38,39] |
| activity | + | we expected that more active individuals would be more successful on our innovative problem-solving tasks [2,6,25,40] |
| neophobia | − | we predicted less neophobic individuals to be more successful on innovative problem-solving tasks [14,36,41] |
| aggressiveness | + | we expected aggressiveness to be positively related to success on our tasks, based on previous work [2,6,14,38,39] |
| risk taking (latency to resume foraging) | − | since risk-taking behaviour is measured via latency to resume foraging after a stressful event [42], we expected that birds with a shorter latency to resume foraging would be better equipped to solve novel problems, so would solve more quickly than birds that took longer to return to feed after experiencing a startling event |
| obstinacy (latency to catch) | + | if obstinacy reflects a bird's persistence to avoid capture [33] and perhaps relates to persistence on novel problems, we predicted that obstinacy would be positively correlated with problem-solving performance |

then separated individually for trials (see 'Behavioural assessment' below). Birds were tested in the same room in which they were housed. For details about the housing timeline, please see figures 1 and 2. Birds had ad libitum access to finch seed and grain mix, vitamin-supplemented water and cuttlebones, at 21°C, approximately 40% humidity, and a 12 h : 12 h light : dark cycle. This study was conducted in accordance with the University of Wyoming Institutional Animal Care and Use Committee (Protocol 20180515SB00307-01).

## 2.2. Behavioural assessment

We conducted one behavioural assessment per day per animal, between 06.00 and 13.45, using the same order of assessments with each bird. Birds were familiar with all cage types used for testing and had habituated to their home cage for at least one week prior to testing. All experiments were video-recorded. See figures 1 and 2 for the detailed experimental timeline

### 2.2.1. Assessment of Dominance Hierarchies (electronic supplementary material, video S1)

Dominance was assessed by recording groups of ten birds in a large flight cage at a single feeder (figure 3a) [40], since monopolization of resources is associated with dominance in zebra finches [33]. The birds had been habituated to their group mates and the feeder for two weeks prior to testing. We

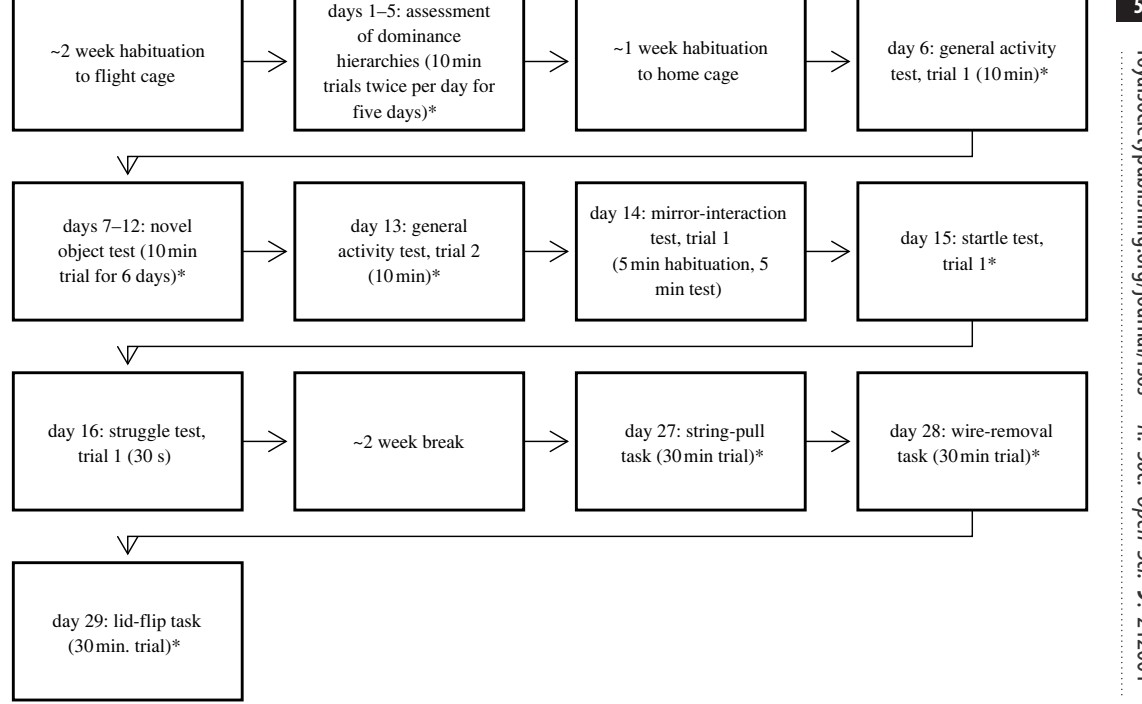

**Figure 1.** Timeline of 2016 behavioural assessments. All assessments took place in the morning. Asterisks indicate tests with food deprivations.

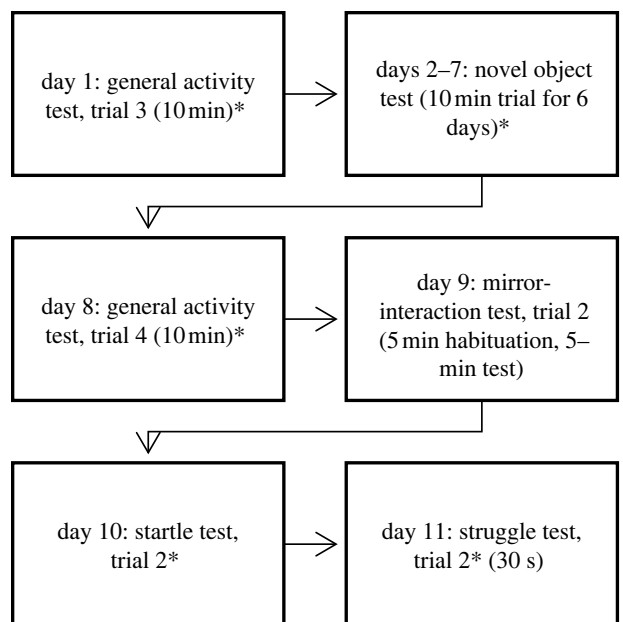

**Figure 2.** Timeline of 2018 behavioural assessments. All assessments took place in the morning. Asterisks indicate tests with food deprivations.

applied pet-safe paint on all of the birds' heads 24 h prior to testing to more easily identify them. We then video-taped birds from above for ten-minute trials that were performed twice per day for five consecutive days. The inter-trial interval for trials on the same day was four hours. One-hour food deprivations began at 07.00 and 12.30, immediately before the start of each trial (at 08.00 and 13.30, respectively). Lights had been on in the room for at least an hour prior to deprivation to allow birds to eat before both morning and afternoon trials. Using the recordings, we counted the number of

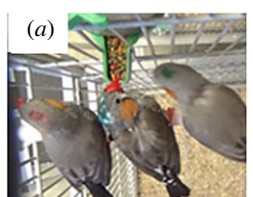 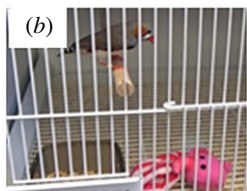 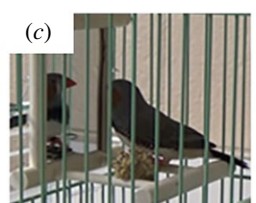 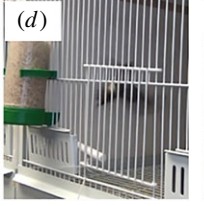 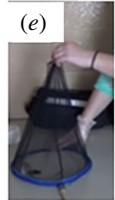

**Figure 3.** Images of behavioural assessments: (*a*) dominance is measured by scoring dominance interactions at a single feeder; (*b*) a novel object is placed near a feeder to test for neophobia; (*c*) a zebra finch is being tested for aggressiveness on the Mirror-interaction Test; (*d*) a zebra finch is startled off of a perch by an experimenter outside the room pulling on a string attached to the perch to assess latency to feed after startle; and (*e*) the number of hops beneath a net are recorded as a measure of obstinacy. We also measured general activity levels (not pictured).

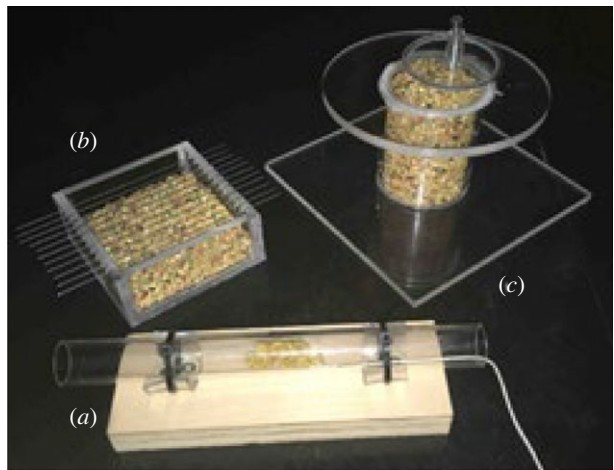

**Figure 4.** Individual problem-solving tasks used in this study: (*a*) string-pull apparatus, (*b*) wire-removal apparatus, and (*c*) lid-flip apparatus. Dimensions were: (*a*) total length of tube: 20.96 cm, tube diameter: 1.91 cm, board width: 5.72 cm, board length: 16.51 cm; (*b*) box length: 8.89 cm, box width: 8.89 cm, wire length: 16.51 cm; (*c*) base width: 15.88 cm, base length: 15.88 cm, circular ledge diameter: 12.70 cm.

interactions in which each bird was involved, defined as physical interaction (peck, bill fence, or fly-at) with another individual on the perch near the single feeder. We also counted how many times each bird remained in its place (as opposed to leaving the perch) following each interaction as its 'number of wins'. Finally, we recorded the number of times each bird perched at the feeder, their time spent at the feeder, and the number of times they fed from the feeder to quantify access to the food resource [44]. Human experimenters were not present in the testing room, and birds were tested in the same groups each trial.

### 2.2.2. General Activity (electronic supplementary material, video S2)

We assessed each bird's intrinsic activity level in the early morning by counting their number of movements in two, ten-minute trials separated by one week [35,36]. Activity is thought to be important in foraging in zebra finches, where more active individuals likely seek out foraging opportunities [45,46]. A movement was defined as flying or hopping between two perches or equivalent distance (at least 23.5 cm). We food-deprived birds for one hour prior to testing to increase motivation [31]. Birds were tested in their home cage. To test birds by themselves while reducing stress of isolation, birds were visually, but not acoustically, separated from a same-sex individual in the home cage by an opaque barrier (i.e. a bird was in half of the home cage) (home cage dimensions: 51.44 cm high × 49.53 cm wide × 50.17 cm deep). Human experimenters were not present in the testing room.

### 2.2.3. Novel object test (electronic supplementary material, video S3)

To test for object neophobia, which we assumed would capture zebra finches' reaction to novelty [44], we placed a novel object, such as a small stuffed animal or ball, within 15 cm of the only feeder in the cage

(figure 3b) (electronic supplementary material, figure S1). Following [33,44], we recorded latency to feed from the feeder, and how many times each bird fed in the 10-min trial. We also recorded latency to perch near the feeder, the number of times the birds perched near the feeder, and the number of times the birds fed. Birds were tested six times (i.e. once per day for 6 days) with a new object each day. Each bird received the novel objects in the same order because birds may habituate to novelty over time, and we required a consistent presentation order so that we could compare the response of the individual birds to the same objects. To determine if a significantly different latency to feed is due to the presence of the object and not differences in general activity, we compared mean responses across all 6 days of testing in Trial 1 (no object) to those in Trial 2 (with object), as well as those in Trial 2 (with object) compared to those in Trial 3 (no object) [39]. Between each trial there was a 2 min inter-trial interval, and we used approximately the same number of seeds in the feeder in each trial. The objects were each placed near the feeder but did not prevent the animal from being able to feed [39]. Birds were tested alone in half of their home cage, and were food-deprived for one hour prior to testing to increase motivation [31]. Human experimenters were not present in the testing room.

### 2.2.4. Mirror-interaction test (electronic supplementary material, video S4)

To test for aggressiveness we measured bird responses to their reflection in a mirror (following [44]). We moved each bird from their home cage to a small cage (24.13 cm long × 17.15 cm wide × 21.59 cm high) with a paper-covered mirror and millet placed just in front of the mirror with a binder clip (see figure 3c) for a five-minute habituation period, during which we did not quantify behaviour. The purpose of the millet, a high-value food reward, was to help entice the birds towards the mirror to see their reflection. We then uncovered the mirror, which always still had millet remaining in front of it, and counted each bird's interactions (defined as landing on the mirror or its perch) with their reflection for five minutes, without any prior food deprivation [30]. Some trials were ended up to one minute short due to technical problems, so the number of interactions were converted to a rate to keep the measure standardized across all individuals. Human experimenters were not present in the testing room.

### 2.2.5. Startle test (electronic supplementary material, video S5)

We measured risk-taking behaviour as the latency (in seconds) for an individual to return to feed after being startled [33,42]. Here we assumed latency to feed after a startle event reflected an individual's ability to cope with an unexpected change or startle [42]. To startle each bird, the experimenter left the test room and watched the bird on a video camera outside. Once the subject perched on a branch and fed from a food dish, the experimenter shook that branch once or twice from outside of the testing room via a string until the bird flew away (figure 3d). Birds were tested by themselves in half of their home cage. Human experimenters were not present in the testing room, and we food-deprived birds for one hour prior to testing to increase motivation [31].

### 2.2.6. Struggle test (electronic supplementary material, video S6)

Obstinacy, also referred to as docility [11–13,47,48], is ecologically relevant in terms of intraspecific competition, or by reflecting escape behaviour when faced with a predator, and it is thought to be a meaningful trait to consider in zebra finch personality research [31]. To test for obstinacy, we recorded the time it took to catch each bird in half of its home cage (where it was separated from its cage-mate), as well as each bird's number of hops (i.e. escape attempts) while under a net in a 30 s time period, without any prior food deprivation [33] (figure 3e). An experienced experimenter attempted to catch a bird alone in its home cage by inserting one hand and following the bird. Once the experimenter caught the bird, the experimenter placed it beneath a net and began the timer for the escape attempt portion of the test. The same experimenters regularly handled the birds (at least once biweekly for health checks or for different testing procedures).

We carried out the first round of tests in 2016. We then carried out the tests (except for the Assessment of Dominance Hierarchies) in 2018 to assess repeatability of measures over two years. We chose this time interval to avoid habituation in these tests over the short term and to assess whether they were repeatable over a considerably longer portion of the zebra finch lifespan [29] (1.3–5 years in the wild and 5–9 years in the laboratory (reviewed in [24]). We assessed repeatability of Assessment of Dominance Hierarchies measures over multiple trials in 2016 (i.e. short term). We could not repeat it in 2018 (i.e. across years) due to birds being housed with a mate in 2018. The Startle, Struggle and Mirror-interaction tests

included only one trial in 2016 and one trial in 2018 in order to avoid habituation in such a short time period (i.e. within years) to these tests in particular because the birds could have learned that the potential threat (i.e. a human trying to catch them, a startle off of the perch, and their reflection in the mirror) is not dangerous. In the other tests (Novel Object and General Activity), we used very different objects to avoid habituation and measure their activity levels, which does not involve substantial experimental manipulation. One experimenter conducted the Struggle Test trials in 2016, and a second experimenter (with similar skill in catching birds) conduced the trials in 2018.

## 2.3. Problem-solving tasks

Approximately two weeks after the 2016 behavioural assessments, we conducted three problem-solving tests that required the birds to move a part of a transparent apparatus to reach a food reward. These tests included a String-pull Test, a Wire-removal Test, and a Lid-flip Test (all adapted from [33]) (figure 4), in that order. Birds received one task per day across three consecutive days, alone in half of their home cage. Prior to testing around noon each day, the birds were food-deprived for 4 hours, during which they habituated to an empty test apparatus [43]. Immediately following the four-hour deprivation/ habituation, the test trial for the day began and lasted for 30 min.

String-pull test (electronic supplementary material, video S7): Birds solved by pulling on a string that was attached to a food reward (millet) inside of a clear tube (figure 4a).

Wire-removal test (electronic supplementary material, video S8): For this task, there were several wires that crossed the top of a tray with a food reward in the bottom (figure 4b). To solve the puzzle, a bird had to pull a wire out of the side of the box in order to reach seed mix.

Lid-flip test (electronic supplementary material, video S9): There was a food reward inside a clear cylinder that was standing on end, covered by a transparent lid in this test (figure 4c). A bird had to move the lid off of the cylinder in order to gain access to seed mix [43].

Birds received one trial per task, during which they could make as many attempts as they wanted, as we were measuring ability to innovate a solution to a novel task. All of the tasks required that the birds inhibit an inherent tendency to peck directly at the seed through the Plexiglas tube, tray or column. We assume that all three tasks likely relied on similar cognitive mechanisms but required somewhat different motor patterns to solve (i.e. peck or pull/slide).

## 2.4. Statistical analysis

To address our first question about whether response measures from behavioural assessments were repeatable among individuals [2,3], we calculated the intra-class correlation coefficient using the 'rptR' package in R, for both short-term repeatability (across trials in 2016) (see figure 1 for details on test–retest intervals) and long-term repeatability [41]. After examining residuals for normality, where necessary we $\log(x+1)$-transformed latency responses and used Gaussian distributions. We assumed Poisson distributions for count responses and for (rounded) rates of interactions with the mirror and added an observation-level random effect to account for overdispersion [41]. For tests run multiple times over the short term or the long term (but not both) we included Trial (or Day 1–6 for Novel Object Test) as a fixed effect and ID as a random effect to test for repeatability. For tests run multiple times over the short term and the long term, we included Trial (or Day 1–6 for Novel Object Test) and Year as fixed effects, and ID and 'Series' (a combination of Year and ID) as random effects (following [49]). Only trial 2 responses from the Novel Object Test (the trial with the novel object present) were included in models. We examined 95% confidence intervals of repeatability estimates between females and males to test for sex differences in repeatability [50]. We retained all long-term repeatable personality measures, as well as short-term repeatable measures from the Assessment of Dominance Hierarchies. We conservatively excluded measures from the same test that were correlated ($r > 0.45$) after a Bonferroni correction (table 2) and checked that the measures were not correlated across tests, using a Pearson correlation and averages responses across trials for each individual.

We next examined which remaining traits (as averages across trials) were associated with problem-solving performance in terms of success in solving each task and average latency to solve the tasks (to test our second question) using regression: we used logistic generalized linear models (GLMs) to determine which of the remaining personality measures were correlated with whether or not birds solved each task, which we tested separately in case the tasks were measuring slightly different aspects of innovative performance. We also ran Gaussian linear models to relate the personality measures with average latency to solve the tasks. Lastly, we used Poisson generalized linear models

**Table 2.** All measures recorded in each year of the study and notes about which measures were excluded for analyses. Asterisk indicates significant correlation after Bonferroni correction.

| test | measure | 2016 | 2018 | reason not measured in 2018 (if applicable) | reason for exclusion from analyses (if applicable) |
|------|---------|------|------|---------------------------------------------|-----------------------------------------------------|
| activity | no. of movements | x | x | | |
| struggle | no. of hops in 30 s | x | x | | |
| | time to catch | x | x | | not repeatable |
| reaction to startle | latency to feed | x | x | | not repeatable |
| mirror interactions | no. of hard pecks | x | | correlated (*) with mirror rate ($r = 0.707$, $p < 0.0001$) | |
| | no. of soft pecks | x | | correlated (*) with hard pecks ($r = 0.455$, $p = 0.003$) | |
| | no. of mirror interactions | x | | correlated (*) with mirror rate ($r = 1$, $p < 0.0001$) | |
| | rate of interactions | x | x | | |
| novel object | average number of times near feeder | x | | correlated (*) with latency to perch ($r = -0.598$, $p < 0.0001$) | |
| | average number of times feeding | x | | correlated (*) with latency to feed ($r = -0.640$, $p < 0.001$) | |
| | average latency to perch at feeder | x | | correlated (*) with latency to feed ($r = 0.539$, $p < 0.001$) | |
| | average latency to feed | x | x | | |
| dominance | average number of interactions | x | x | | correlated (*) with average wins ($r = 0.814$, $p < 0.0001$) |
| | average time at feeder | x | x | | correlated (*) with times at feeder ($r = 0.803$, $p < 0.0001$) |
| | average number of wins | x | x | | |
| | average number of times feeding | x | x | | correlated (*) with times at feeder ($r = 0.998$, $p < 0.0001$) |
| | average number of times at feeder | x | x | | |

**Table 3.** Long-term repeatability estimates ($R$), SEs, low and high 95% CI limits, and associated $p$-values for behavioural tests across 2016 and 2018, adjusted for trial and year. Shading separates personality assessments. Bold values indicate significant ($R > 0.20$ and $p < 0.05$) repeatability estimates.

| behaviour | test | $R$ | SE | low CI | high CI | $p$ | $n$ |
|---|---|---|---|---|---|---|---|
| latency to feed | novel object test | **0.162** | **0.064** | **0** | **0.295** | **<0.0001** | **36** |
| rate of interactions | mirror-interaction test | 0.266 | 0 | 0.266 | 0.266 | 0.062 | 36 |
| latency to feed | startle test | 0.073 | 0.140 | 0 | 0.462 | 0.364 | 36 |
| no. of hops | struggle test | **0.224** | **0.134** | **0** | **0.511** | **0.042** | **36** |
| time to catch | struggle test | 0.227 | 0.145 | 0 | 0.529 | 0.084 | 36 |
| no. of movements | general activity test | 0.080 | 0.099 | 0 | 0.337 | 0.225 | 36 |

**Table 4.** Short-term repeatability estimates ($R$), SEs, low and high 95% CI limits, and associated $p$-values for behavioural tests across multiple trials in 2016, adjusted for trial. Struggle, Startle, and Mirror-interaction Tests were only conducted once in 2016 and are therefore not shown in this table. Shading separates personality assessments. Bold values indicate significant ($R > 0.20$ and $p < 0.05$) repeatability estimates.

| behaviour | test | $R$ | SE | low CI | high CI | $p$ | $n$ |
|---|---|---|---|---|---|---|---|
| **latency to perch near** | **novel object test** | **0.469** | **0.075** | **0.318** | **0.603** | **<0.001** | **41** |
| **latency to feed** | **novel object test** | **0.165** | **0.068** | **0.022** | **0.295** | **<0.001** | **41** |
| times at feeder | novel object test | 0.122 | 0.044 | 0.032 | 0.21 | <0.001 | 41 |
| times at feeder | assessment of dominance hierarchies | **0.989** | **0.251** | **0.488** | **1.493** | **<0.001** | **41** |
| number of interactions | assessment of dominance hierarchies | **0.262** | **0.05** | **0.169** | **0.365** | **<0.001** | **41** |
| time feeding | assessment of dominance hierarchies | **0.460** | **0.066** | **0.329** | **0.583** | **<0.0001** | **41** |
| no. of wins | assessment of dominance hierarchies | **0.886** | **0.273** | **0.474** | **1.512** | **<0.0001** | **41** |
| no. of times feeding | assessment of dominance hierarchies | **0.276** | **0.063** | **0.149** | **0.400** | **<0.0001** | **41** |
| no. of movements | general activity test | 0.235 | 0.135 | 0.105 | 0.416 | 0.010 | 41 |

(after checking for overdispersion) to relate each personality measure with the total number of puzzles solved. The GLMs of solving success for each of the tasks included those behaviours that were repeatable over the long-term (2 years): average latency to feed (Novel Object Test) and average number of hops (Struggle Test) (table 3). We also included average number of times at feeder (Assessment of Dominance Hierarchies) and average number of wins (Assessment of Dominance Hierarchies), which were repeatable over the short term (about two weeks) (table 4) but not tested over the long term. We tested for variance inflation factors to exclude the possibility of covariance between personality traits influencing the model outcomes and used Akaike's information criterion (AIC) for our model selection. AICc values were calculated using the 'MuMIn' package in R [49]. To determine if the three tasks were independent measures of cognitive ability, we used GLMs to assess repeatability of success (binomial) on the tasks and latency to solve the tasks as rounded whole numbers (Poisson), where Task was fixed effect and ID was a random effect. We conducted *post hoc* GLMs to determine if the interaction of sex and each personality trait affected problem-solving performance measures (whether or not birds solved each task, average latency to solve the tasks, and number of tasks solved). All analyses were conducted in R [50].

# 3. Results

## 3.1. Behavioural assessments

Most measures from the behavioural assessments were repeatable, or consistent in the short term (across trials in 2016), but the number of times at the feeder (Novel Object Test) was not repeatable across trials (table 4). Of the measures repeated in 2018 (i.e. the long term), two measures were repeatable: latency to feed (Novel Object Test) and number of hops (Struggle Test) (table 3). Wilcoxon rank tests indicated a significant difference between Novel Object Trial 1 and Trial 2 responses, where Trial 2 had a higher latency to feed ($p < 0.001$), as well as between Trial 2 and Trial 3 responses, where Trial 2 again had a higher latency to feed ($p < 0.001$).

We also examined sex differences in repeatable traits retained for analyses on their connection to innovative performance (see Statistical analysis section). Behaviours that were repeatable over the long term, latency to feed (Novel Object Test) (females: $r = 0.148$, CI: [0, 0.48]; males: 0.181 [0.022, 0.375]) and number of hops (Struggle Test) (females: 0.448 [0, 0.796], males: 0.237 [0, 0.648]), did not differ in repeatability by sex, as indicated by overlapping confidence intervals. Dominance measures (Assessment of Dominance Hierarchies; repeatable over the short term but not tested over the long term) also did not differ in repeatability by sex (times at feeder: females: 0.604 [0.318,0.789], males: 0.544 [0.244,0.741]). Number of wins (Assessment of Dominance Hierarchies) showed nonoverlapping confidence intervals (females: 0.576 [0.208,0.779], males: 0.145 [0,0.149]), but male repeatability was low (less than 0.20) and not significant ($p = 0.06$).

## 3.2. Problem-solving performance

There was pronounced inter-individual variation in success on the three problem-solving tasks: 3 birds did not solve any tasks, 11 birds solved one task, 19 birds solved two tasks and 8 birds solved all three tasks. 56% of birds (23/41) solved the String-pull Test, 80% (32/40) solved the Wire-removal Test, and 50% (20/40) birds solved the Lid-flip Test. The fastest latency to solve was just a few seconds after placing the apparatus in the home cage (technically before the start of the trial) and was assigned a value of zero seconds. The slowest latency to solve was 28.23 min (on the Lid-flip Test). Non-solvers (including birds who approached but did not touch the apparatus) were assigned a ceiling value of 1801 s for latency to solve. There was a trend that latency to solve was repeatable across tasks, but this was not significant ($r = 0.387$, $n = 41$, $p = 0.052$). Whether or not birds solved the tasks (0/1) was not repeatable across tasks ($r = 0.000$, $n = 41$, $p = 0.440$).

## 3.3. Personality–cognition associations

See the Statistical analysis section for retained personality traits. Our top model for success on the String-pull Test included a negative effect of average latency to feed (Novel Object Test) (figure 5; table 5) but the effect was small (electronic supplementary material, table S1). Our top model for success on the Wire-removal Test included a small, negative effect of average number of times at feeder (Assessment of Dominance Hierarchies) (figure 5; table 5; electronic supplementary material, table S1). Our top model for success on the Lid-flip Test included a negative effect of average number of wins (Assessment of Dominance Hierarchies), a negative effect of average number of times at feeder (Assessment of Dominance Hierarchies), and negative effect of average latency to feed (Novel Object Test) (figure 5; table 5; electronic supplementary material, table S1). Average latency to solve the three tasks was positively related to average number of times at feeder (Assessment of Dominance Hierarchies) and negatively related to average number of hops (Struggle Test) (table 5; electronic supplementary material, table S1). The total number of tasks solved was negatively related to average number of times at the feeder (Assessment of Dominance Hierarchies), average number of wins (Assessment of Dominance Hierarchies) and average latency to feed (Novel Object Test) (table 5; electronic supplementary material, table S1). We did not find a significant effect of the interaction between sex and each personality trait in our *post hoc* GLMs relating personality to success on each task, average latency to solve the tasks or the total number of tasks solved.

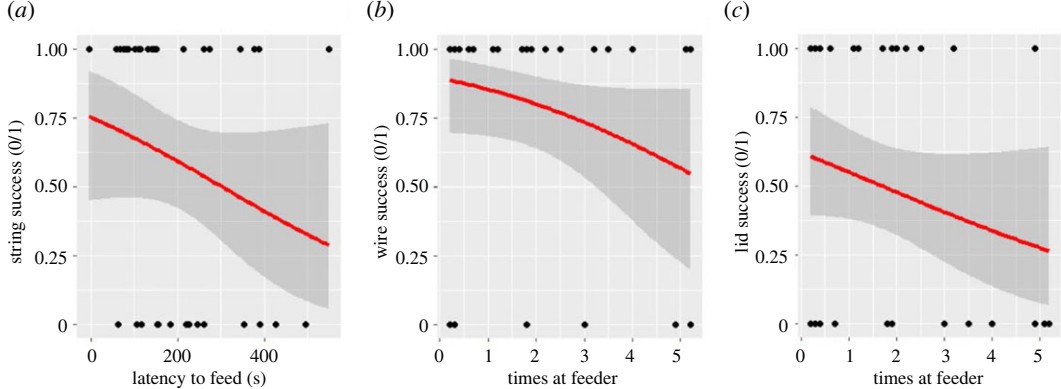

**Figure 5.** (*a*) Birds who were quicker to feed next to a novel object were more likely to solve the String-pull Test. (*b*) Birds who fed fewer times at the single feeder were more likely to solve the Wire-removal Test. (*c*) Birds who fed fewer times at the single feeder were more likely to solve the Lid-flip Test.

## 4. Discussion

Overall, we found evidence of short- and long-term repeatability in several of our behavioural measures, which informs our understanding of the links between personality and innovation. In particular, we found support for repeatable traits (neophobia and obstinacy) over two years (research question 1). We also found that boldness, obstinacy and dominance were related to innovative performance (research question 2).

Addressing our first research question about repeatability of behaviour, we found evidence of consistent individual differences in intrinsic activity, dominance, neophobia, and obstinacy that were repeatable across days in this population, suggesting they can be considered personality traits and supporting previous findings of repeatability in these traits in zebra finches [33]. Our levels of repeatability support previous findings of repeatability levels of traits in other animals as well (reviewed in [23]). We found that two traits, namely neophobia and obstinacy, were repeatable over two years, a substantial portion of a wild zebra finch's average lifespan [29]. Although neophobia has been well studied (e.g. [23]), to our knowledge, obstinacy (intensity of struggle behaviour) has not previously been measured over such a long time period. The repeatability we observed in this understudied trait suggests it could be important for understanding animal personalities in birds. We, therefore, emphasize a previous call [31] for animal personality research to broaden the range of traits measured (including obstinacy) in order to accurately capture the complexity of animal personalities.

We consider several possible explanations for why some traits were not repeatable over two years in this study. For the Struggle Test, for example, the time required to catch a bird in its home cage was not repeatable over two years, although the number of escape attempts beneath a net was repeatable. Despite our best efforts to maintain consistency in the experimenter's catching ability, it is possible that human variation between years influenced the repeatability of time to catch between the two years. A *post hoc* nonparametric Wilcoxon test indicated a significantly lower time to catch on average in 2018 ($\bar{x}$ = 14.73 s) compared to 2016 ($\bar{x}$ = 21.48 s) ($W$ = 639, $n$ = 30, $p$ = 0.005), which could indicate an effect of age or habituation/experience for the birds, or experience on the part of the experimenter. For the Struggle Test, the observed variation in repeatability among different measures of the same trait illustrates an inherent challenge to studies of animal personality, namely the difficulty of establishing a single test to capture a personality trait [29]. Other studies of obstinacy in birds found that obstinacy was not repeatable [51] and unrelated to any other measured personality trait [31]. Next, latency to feed after being startled off of a perch in the Startle Test also did not seem to be consistent across two years. Interestingly, we based this assessment on the startle test used in zebra finches [33], where the authors found that the latency to resume feeding after startle was repeatable across one week, suggesting an effect of having a short test–retest interval [33], which corresponds with work in other species [28]. Since we did not retest in a short interval, we do not know if we would have found significant repeatability in the short term, so perhaps the difference is due to another difference between the studies, such as the birds tested, the methods, or something other than risk-taking in the birds. General activity was not repeatable over the long term, which is consistent with another study on zebra finch personality that found activity to be repeatable only in subadults and not across life

**Table 5.** Summary table for models generated with Δ AICc less than or equal to two. Top models are indicated by an asterisk (*).

| model | type (distribution) | d.f. | $R^2$ | AICc | Δ AICc | Akaike weight |
|---|---|---|---|---|---|---|
| string solve | | | | | | |
| * string ~ average latency to feed (novel object test) | GLM (binomial) | 2 | 0.171 | 51.3 | 0 | 0.675 |
| wire solve | | | | | | |
| * wire ~ average number of times at feeder (assessment of dominance hierarchies) | GLM (binomial) | 2 | 0.083 | 41.8 | 0 | 0.325 |
| wire ~ average number of times at feeder (assessment of dominance hierarchies) + average latency to feed (novel object test) | GLM (binomial) | 3 | 0.088 | 42.0 | 0.19 | 0.295 |
| wire ~ average latency to feed (novel object test) + average number of times at feeder (assessment of dominance hierarchies) + average number of hops (struggle test) | GLM (binomial) | | | | | |
| wire ~ 1 | GLM (binomial) | 1 | 0 | 42.1 | 0.35 | 0.273 |
| lid solve | | | | | | |
| * lid ~ average number of wins (assessment of dominance hierarchies) + average number of times at feeder (assessment of dominance hierarchies) + average latency to feed (novel object test) | GLM (binomial) | 4 | 0.270 | 50.8 | 0 | 0.713 |
| average latency to solve | | | | | | |
| * average latency to solve ~ average number of times at feeder (assessment of dominance hierarchies) + average number of hops (struggle test) | GLM (Gaussian) | 4 | 0.437 | 483.6 | 0 | 0.736 |
| no. of puzzles solved | | | | | | |
| * no. of puzzles solved ~ average latency to feed (novel object test) + average number of wins (assessment of dominance hierarchies) + average number of times at feeder (assessment of dominance hierarchies) | GLM (Poisson) | 4 | 0.321 | 104.8 | 0.00 | 0.789 |

stages [23]. More research is needed to examine the developmental plasticity of activity in general [29]. Finally, in the Mirror-interaction Test, the rate of interactions with the mirror was not repeatable over two years. Previous work has found evidence of consistent aggression in zebra finches across two trials with a one-week inter-trial interval [30], whereas others have only tested for aggression in a single trial [52]. This, too, suggests that the test was not repeatable in our study because of our long test–retest period, but we could not examine that empirically. Additionally, we used millet to entice birds toward the mirror (following [27]), and so food motivation could have affected their responses and should be considered in future studies. We advise future studies of mirror-interactions to conduct testing in the absence of food.

That some—but not all—measures of the behavioural assessments were repeatable over the short and long term supports previous findings that repeatability is susceptible to biases of interval duration, where measures recorded closer to another in time tend to be more repeatable than those measured over longer intervals [31]. It is also possible that there was some other difference in study subjects or methods. For example, a change in physiology, ageing, or adjusting to new housing conditions could affect behavioural variation and change in individuals' behaviour over time. Indeed, we were unable to measure dominance over the long term because the birds were housed with a mate in 2018; the change in housing could have reduced our long-term repeatability results of all of the tests. We also note that although we carefully chose assessments used in the animal personality literature [9,24,27,30,31,40], not all of them have been validated in an ecological setting, and so it is possible we were not measuring the traits we intended to measure. Ideally future work can examine whether the tests used here that are commonly used in captivity translate to zebra finches' responses to environmental stimuli in the wild, following [53]. It is also likely that the tests used here were influenced by multiple traits simultaneously, or that some traits carry more significance for zebra finch fitness than others [4,54]. Finally, our lack of sex differences in repeatability of personality also supports a recent meta-analysis which did not find widespread sex differences in personality across taxa, including in traits of activity, aggression and boldness in birds [55]. Our study emphasizes that it is critical to investigate multiple behavioural measures of multiple personality traits, across longer time periods, when drawing conclusions regarding the repeatability of behavioural traits.

Beyond measuring repeatability of a suite of traits, we also elucidated links between personality and cognition to address our second research question. We found that bolder individuals were more likely to solve the String-pull Test compared to shyer individuals, but this was a weak effect. This supports our prediction, and findings in other species [15,19,56], that bolder individuals would be more likely to solve innovative problem-solving tasks. It is possible that for the String-pull Test, neophobia was most important because the String-pull Test was the first task the birds received. As such, future work should investigate whether there are patterns with regard to neophobia and task order. Success on the Wire-removal Test was related to average number of times at the single feeder (i.e. a measure of dominance), where less dominant birds were more likely to solve compared to more dominant birds. Thus, we found support for the prediction that less dominant individuals would be more successful, which is consistent with the 'necessity drives innovation' hypothesis that has been supported in other taxa [7,16]. Less dominant zebra finches in this population may need to innovate new ways to access resources compared to their more dominant counterparts [34], but this remains to be confirmed given that the zebra finches tested here were housed with access to multiple feeders and *ad libitum* food outside of test conditions. Similarly, in wild great tits, subordinate individuals adopt alternative foraging strategies to avoid the costs associated with competing for access to food, and are better problem solvers than more dominant individuals [24]. Here, lid-flip success was related to both measures of dominance and latency to feed (i.e. boldness), where bolder, less dominant birds were more successful on this task. Unlike the other tasks that could be solved from the ground, the lid-flip task was solved from above, and many birds solved it by perching on the apparatus. Boldness could be important in approaching and perching on this novel apparatus, and dominance could play a role in ability to gain access to the food, as noted above. It is possible that each task tested a slightly different aspect of innovative problem solving, and required different traits to succeed, so we did not find the same personality measures associated with all problem-solving task performance measures (i.e. they are not part of a cognitive syndrome). In fact, we found that success on the tasks (yes/no) was not repeatable, but we found a strong trend toward repeatable time to solve across tasks. This suggests that the tasks were different from one another in that individuals that excelled in one task did not always solve the others. Additionally, we have some evidence that the tasks were fairly equivalent in difficulty level in that successful individuals solved each task in a comparable amount of time. Alternatively it is possible, though unlikely, that cognitive ability is a fixed trait or that the tasks did not measure cognitive ability [57]. However, more work is needed to identify whether the tasks differed from one another from a cognitive perspective. Future work could also examine whether learning ability on the tasks is repeatable, as has been found in a recent study of zebra finch learning across contexts [58].

We also measured average latency to solve the three problem-solving tasks, and found it was related to dominance and obstinacy: less dominant, but more obstinate birds tended to solve the tasks more quickly, on average. It is possible that more obstinate birds have higher metabolic rates and require more energy, and thus are more food-motivated, or are otherwise more persistent compared to less obstinate birds. The total number of tasks solved was related to dominance and boldness where less

dominant but bolder birds tended to solve a greater number of tasks. Nevertheless, measures of boldness and dominance were related to multiple indicators of success, and are likely important for problem solving in general. We acknowledge, however, that we were limited by our data structure and used only a single measure (mean) per individual instead of a multivariate analytical approach. This approach ignores the error variance around individual mean behaviour, and can lead to spurious significance [59–61].

More generally, our results suggest that some behavioural traits remain consistent and are conserved within individuals over a long time period, while other traits are more plastic over time, confirming results of a previous meta-analysis across vertebrates and invertebrates both in the field and in the laboratory [31]. Differences in repeatability could be due to developmental reasons (e.g. hormones), as has been found recently in dairy cattle entering puberty [62], mean-level population changes in personality [31], or captive living, where there may be less among-individual variation [31]. Also, since repeatability is a population-level measure, it is possible that some individuals did remain consistent while others did not, and so individual-level consistency would be worthwhile to explore (discussed in [29]). We also found that several personality traits were weakly correlated with problem-solving success in a task-dependent manner, suggesting again that the tasks used here measured slightly different aspects of innovative ability and/or required different behavioural traits for success. Variation in ability to solve weakly depended on latency to feed near a novel object (i.e. boldness) on the first and third tasks and on access to food in a social context (i.e. dominance) in the second and third tasks. Our results provide an impetus for further examination of the importance of traits—especially boldness and dominance—for performance on other types of innovative tasks and provide some additional results on the relationship between boldness and innovation [22]. The strongest effect of personality in the present study was on average latency to solve all three of the tasks, where less dominant, more obstinate birds solved fastest. Perhaps less dominant birds have a greater need to innovate so they are not displaced by more dominant birds, and perhaps more obstinate birds are more persistent in their attempts to innovate. Our results suggest, therefore, that the 'necessity drives innovation' hypothesis [20,25,26] applies across repeated tests of innovation, and we recommend future studies integrate measures of obstinacy when assessing innovative performance.

A gap remaining from previous animal personality studies is that they have typically examined the role of one or two personality traits on their own and across short time periods between trials. We have started to fill this gap in our understanding by leveraging a comprehensive suite of personality tests across short (i.e. several days) and long (i.e. 2 years) time periods. We also expanded on the traditional presentation of a single cognitive task by using three tasks in order to more fully capture the association between personality traits and innovative performance. It is important to consider measuring suites of traits for future studies linking personality and innovation, because lesser-studied traits (e.g. obstinacy) could play an important role in innovativeness. Moreover, although boldness and dominance were important for innovation here, not all measures of boldness and dominance were consistent over time, so future work should aim to uncover which measures of personality are most important for innovation and why some traits are more plastic than others. We also encourage future studies to leverage multiple tests of innovation in order to explore the effect of task order on the link between personality and innovation.

Data accessibility. Data and code are available in a Dryad Digital Repository: https://doi.org/10.5061/dryad.gmsbcc2mw [63].

Authors' contributions. L.P.B.: conceptualization, data curation, formal analysis, funding acquisition, investigation, methodology, project administration, visualization, writing—original draft, writing—review and editing; J.L.M.: conceptualization, data curation, funding acquisition, investigation, methodology, writing—review and editing; N.J.B.: conceptualization, methodology, resources, supervision, writing—review and editing; C.N.T.: conceptualization, methodology, resources, supervision, writing—review and editing; S.B.-A.: conceptualization, funding acquisition, methodology, project administration, resources, supervision, writing—review and editing.

All authors gave final approval for publication and agreed to be held accountable for the work performed therein.

Conflict of interest declaration. We declare we have no competing interests.

Funding. We thank the Animal Behaviour Society for funding L.P.B., and NIGMS Wyoming INBRE for funding J.L.M. under grant no. P20GM103432. This material is based upon work supported by the National Science Foundation Graduate Research Fellowship to L.P.B. under grant no. DGE-1747504.

Acknowledgements. We thank three anonymous reviewers and the editor for their suggestions which have strengthened this article. We are very grateful to all of the research assistants who made this project possible, especially Kristina Bartz, Rachel Graham, Stephanie Hart, Karen Hendrick, Phoebe Ramirez and Rachel Ziejka. Many thanks to the

Animal Behaviour and Cognition Lab and the Grindstaff Lab for helpful feedback and discussion, to Steve Devries for apparatus construction, and to Robert Carroll and Jonathan Prather for support and guidance. We thank Jolle Jolles for statistical advice.

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
