## [Peer Review File · Royal Society Open Science]

Review History

RSOS-210413.R0 (Original submission)

Review form: Reviewer 1

Is the manuscript scientifically sound in its present form?

Yes

Are the interpretations and conclusions justified by the results?

Yes

Is the language acceptable?

Yes

Do you have any ethical concerns with this paper?

No

Have you any concerns about statistical analyses in this paper?

No

Recommendation?

Accept with minor revision (please list in comments)

Comments to the Author(s)

The present study assesses links between different personality traits and three different problem-solving tasks in captive zebra finches. Several personality traits predicted success in the problem-solving tasks but some associations were task-specific.

The manuscript is overall well written and addresses a relevant question that has recently received increasing attention; that is, whether personality traits and cognitive abilities are correlated. While the question is not new, the authors are among the firsts to assess such links across different problem-solving tasks.

I have several rather minor comments that I would like to see addressed.

L61-71: I would wish for an explanation as to why it would be important to understand the link between personality and cognition.

L73: Please change to 'problem'.

L83: It says 'typically' - which studies are the exception? Please mention those here.

L85: Please clarify which definition of personality is used. According to the abstract, personality differences are 'consistent differences across time or contexts...', suggesting they can be single traits; here it is suggested that they are 'multidimensional collection of behavioural traits'. Would be 'syndromes' more appropriated here?

L91 onwards: Several studies have already assessed personality traits across longer periods in zebra finches; these should be acknowledged here (e.g. 7 months: David et al. 2012; 3 months: Schuett et al. 2011; 1 year: Wuerz et al. 2015).

L131: I suggest using degree Celsius instead of Fahrenheit.

L 155: Could you give a more complete list of physical interactions that were used, please?

L193: Were birds tested in the same groups across the years?

L223: It seems I do not have access to video 7-9.

L259: Are the patterns / repeatability values similar if only the data of the first trial 2016 and 2018 are used? As repeatabilities in the short term are not that high, it seems more appropriate not to use the means.

Analyses/Results: Potential sex-differences in patterns have not been assessed, even though some studies have shown that repeatabilities of personality traits significantly differ between males and females and that links between personality and cognitive abilities can be sex-specific (for latter, see also intro of current study). Even though the sample size is not huge, I suggest considering such potential sex differences in the analyses.

L265: Were model assumptions checked and met? Usually, latencies do not follow Gaussian error structures. Please clarify in the text.

Line 298-304: It is unclear if those are the maximal models or reduced models; if the first, it seems more appropriate to add this info to the stats section.

L304-313: I suggest adding the direction of effect, especially for those results that are not summarised in Fig. 3 captions.

Discussion: the section on repeatability of behaviour is quite long and could be shortened considerably.

SI Document 1: I suggest adding this information to Fig 2 and deleting this document here.

References:

David, M., Auclair, Y., & Cezilly, F. (2012). Assessing Short- and Long-Term Repeatability and Stability of Personality in Captive Zebra Finches Using Longitudinal Data. *Ethology*, 118(10), 932-942. doi: 10.1111/j.1439-0310.2012.02085.x

Schuett, W., Dall, S. R. X., & Royle, N. J. (2011). Pairs of zebra finches with similar 'personalities' make better parents. *Animal Behaviour*, 81(3), 609-618. doi: 10.1016/j.anbehav.2010.12.006

Wuerz, Y., & Krüger, O. (2015). Personality over ontogeny in zebra finches: long-term repeatable traits but unstable behavioural syndromes. *Frontiers in Zoology*, 12(1), 1-14.

Review form: Reviewer 2

Is the manuscript scientifically sound in its present form?

No

Are the interpretations and conclusions justified by the results?

No

Is the language acceptable?

Yes

Do you have any ethical concerns with this paper?

No

Have you any concerns about statistical analyses in this paper?

Yes

Recommendation?

Reject

Comments to the Author(s)

Please see all comments in the attached file (see Appendix A).

Review form: Reviewer 3

Is the manuscript scientifically sound in its present form?

No

Are the interpretations and conclusions justified by the results?

No

Is the language acceptable?

Yes

Do you have any ethical concerns with this paper?

No

Have you any concerns about statistical analyses in this paper?

No

Recommendation?

Major revision is needed (please make suggestions in comments)

Comments to the Author(s)

This study examines 1) the long-term repeatability of personality traits, and 2) whether multiple cognitive tests correlate with personality measures. A battery of personality tests (dominance test, general activity measurement, novel object test, startle test, struggle test, mirror-interaction test) and a battery of problem-solving tests (string-pull test, wire-removal test, and a lid-flip test) were conducted on 41 captive zebra finches. Personality tests were repeated after a short interval, and partly repeated after two years. Results showed that most variables measured in the personality tests were consistent over a two years period. The success in problem-solving tests was best predicted by dominance and boldness, with less dominant but bolder birds more likely to solve a higher number of tests.

I think this is a very nice and thorough study, with the potential to advance scientific knowledge in the field. However the presentation of this work will need extensive revisions. Unfortunately, I think that the manuscript, data analyses and result interpretation all need further work before being ready for publication. The overall aim of the study does not become apparent yet, the take away message remains unclear and diffuse, the story still needs shaping, and conclusions seem at least disconnected from the results. In short, I do not think the authors do justice to the impressive amount of effort that went into this data collection.

General comments

From the way it is currently presented, it is unclear whether the authors consider this a methodological study, highlighting potential and difficulties of measuring multiple personality traits in relation to cognition, or actually a study investigating this connection and its meaning for cognitive and behavioural evolution and research. The two levels are constantly mixed. If this is a study whose results are intended to contribute to the understanding of individual variation in cognition (and I think it definitely has the potential to do this), there is no clear reasoning on why it was attempted, and what the obtained results mean in a biological sense. The results illustrate that there is a connection between some personality measures and problem solving, but why is the presence of this connection considered proof that running multiple tests is important? I do agree that this is what results show, and also that this methodological approach is important, but I do not see an evident connection in how this is presented here. It would be good if the authors clarified what they think is the main point of this paper, a validation of a methodological approach or the biological relevance of this link. This will help to focus more clearly on the aspects they want to highlight, and to create a hierarchical order of importance of their findings in the narrative that will be clear to readers as well.

My second issue concerns the claim that multiple cognitive traits were assessed in combination with multiple measures of personality. The problem-solving tests were repeated (great!), and even though it is not clear if a similar assessment was carried out in 2018, this is a very important aspect that has all my approval. However, while the personality tests indeed address multiple aspects of animal personality, the three cognitive tasks are variations of the same test, a removal of something to access the food reward visible through a transparent barrier. The three tasks were targeting the same cognitive ability, and relied on similar cognitive mechanisms (L234). The variation was in the non-cognitive aspects such as motor actions required to solve them.

The papers that the authors cite (e.g. Griffin et al., 2015) in support of this approach, actually recommend testing multiple cognitive traits along with their repeatability and other personality traits. While the authors often write “three problem-solving tests”, mentions of repeated tests in the context of the need for multiple cognitive assessments are misleading. My recommendation is that the authors clarify that the multiple traits measured here only refer to personality (to avoid overselling) and, more importantly, that they explain why they think is important multiple measures of personality in connection with a single, albeit repeated, cognitive task. At the moment the reader is left with the impression that the main reason is that some measures of personality correlate with the problem-solving performance and others are not, without going any deeper into the implications of this connection.

Another aspect that in my opinion requires revision is that, currently, the rationale for this study is barely explained. The authors cite the meta-analysis that found that while personality and

cognition are related, the direction of this relationship is not the same across species, or tasks etc. From the way this is presented, I wonder if we should expect any kind of biological relationship to be so universal and unidirectional, and why the species- or task-dependence is considered problematic. Personally, I think it is great that further studies are conducted on the topic – I just could not understand what are the authors ideas on this, though, as they are not presented in the text beyond the simple statements that the relationship not always goes in the same direction and that, for this reason (?), multiple (personality) traits should be considered. This intellectual contribution in contextualising findings is, in my opinion, quite as necessary as robust results themselves.

Finally, I think that the thoroughness with which the personality tests were conducted should extend to the analysis of the data derived from them. The various tests yielded a wealth of behavioural observations on several variables – but of course not every single variable measured in, for example, a neophobia test is an indicator of neophobia, and most likely expresses a combination of various traits. Currently, all variables are presented as carrying the same significance and being indicators of only the personality trait after which the corresponding test is named. However, the study of animal personality requires a more critical approach towards the measured variables, and a careful interpretation and evaluation of the observed responses. Have these tests been validated in an ecological setting before concluding that they actually measure boldness, aggressiveness, etc.? If not, what does it tell us that an individual approaching a mirror has a better chance of solving a foraging task? One test can simultaneously be influenced by and thus measure two or more personality traits. Although a test that directly measures a targeted trait may be a desirable goal, in reality a test will likely be influenced by multiple traits at the same time (Carter et al., 2013; Réale et al., 2007). A critical re-evaluation of the variable presented as personality traits is required before this work could be considered ready for publication. On this, I recommend the review by Carter et al. (2013, *Biol. Rev.* (2013), 88, pp. 465–475. 465, doi: 10.1111/brv.12007) as a very valuable starting point to start the revision process.

Specific comments

L36-39: no connection between the first two sentences of the abstract. Why is it important to examine personality and cognition in the same study? I think this info should come before the methodological approaches used in the past, to place this study in the context of current knowledge.

L45-49: From this conclusion it seems that the aim of the study is to support the recommendation of conducting multiple assessments of personality, but it does not address why. Why is it important for the study of cognition to have robust proof that personality is linked to cognition? And why is it important to know under which circumstances and how specific personality traits affect performance in cognitive tests?

L60-116: In its current form, the introduction is a mixture of unconnected statements about personality being important for fitness, having a non-linear relationship with cognition, and methodological approaches. Considering that personality is the consistent among-individual variation in behavioural responses, I would have expected some mentions on why variation in cognitive performance is important, why we should be concerned with it, and why should we consider the other behavioural aspect of the individuals under study to properly assess the behavioural performance we use to infer cognitive processes and abilities. Instead there are a lot of disconnected sentences that do not reach the heart of the problem, in my opinion, but keep mixing the methodological level with the biological significance of the topics addressed. Of course, this does not detract from the merits of the study, but makes it much harder to appreciate them. I recommend an extensive revision of this section.

L61-71: the introductory paragraph does not seem to lead clearly to the research question. Why is this link considered relevant? And what were the basis to expect that the relationship between and cognition should be a universal constant, where most behavioural and cognitive aspects are indeed sex-, condition-, age-, motivation-, and context-dependent (L68-70)?

L70-71: please explain briefly the choice of this particular aspect of cognition, also in reference to the ongoing debate on whether problem-solving and innovation are indicators of cognitive ability

(e.g. Chen et al., 2020; Camacho-Alpízar et al., 2020). While I personally support this view, I believe that in this context the debate should at least be acknowledged.

L73-74: please explain in what way innovation is crucial to find new resources, it does not transpire clearly from this sentence (“Innovation, the ability to solve novel problems or to find new solutions to an old problems [8], is crucial for locating and using novel resources in a changing environment”) and is not mentioned elsewhere in the intro.

L72-77: these three sentences are a good example of the issues I find with the presentation of this study. The frequent oscillation between biological relevance of the topic addressed here – the link between personality and innovation, which is often mentioned but never addressed in depth – and methodological aspects. I believe that if the authors find the connection they investigate meaningful in a biological sense, it should be addressed beyond the mention of the one meta-analyses that found that the direction of the relationship varies. And the significance of this relation should be kept separate from the methodological considerations.

L76-77: please provide support for these statements, as they refer to existing knowledge and previous studies.

L77-83: after describing the variation in results, I suggest making it explicit why this is considered problematic. Why would it be good to have a one-way expectation for the link between personality and problem-solving performance? And do the authors think that given the diversity of morphology, environments, social structure in the listed taxa, such a one-way relationship is really possible?

L89-90: I disagree: personality might affect the performance, the way in which a cognitive task is approached and solved, not the ability itself.

L91-93: And what is the significance of having some personality measures repeatable after an interval of 2 years? What does it mean for zebra finches personality, its evolution, its trade-off with plasticity? And what methodological conclusions can be derived from knowing that zebra finches maintain some of their behavioural responses consistent throughout a significant part of their life?

L101: the fact that a study has not been conducted in a certain species is not a sufficient reason to justify one. Please provide reasons why you consider this species a suitable model to address your question, and why it should provide answers that are valid across taxa, as it seems implied in the introduction and abstract in their current form.

L104-107: “Individuals capable of accessing food resources (such as more dominant individuals), especially when food is ephemeral, may possess an advantage over other individuals to expand the amount or type of food available to them. We filled this gap in knowledge by [...]” I see no gap in knowledge in the previous sentence, and the statement does not really explain the study, especially since foraging or fitness consequences are not assessed.

L112-113: Why is it important to have multiple measures of both cognitive performance and personality? What does that add to both fields? A reader should not be forced to Griffin et al. (2015) to start to have answers on this. Please revise the introduction so that the relevance of the study is clear, even to reader who are not familiar with the studies you cite.

Table 1: I am unclear how some of these predictions were developed. If the relationship between personality and innovation varies across species and no previous study was conducted in zebra finches, how did you choose which studies to rely on to inform the predictions (e.g. on dominance?).

There is a single prediction clumping together activity, neophobia and aggressiveness; are these traits considered interchangeable? Given the ecological meaning of each, and how very distinct from one another they are, I found it difficult to understand why they are considered together in here – unless this was done in light of the obtained results? If that is the case, please revise, as predictions should arise well before knowing the study outcome.

The predictions on risk-taking are mixing the methodological approach (how this trait is measured – in a fraction of studies) with its significance and potential to affect problem-solving performance and innovation propensity. Lastly, I have never come across the term “obstinacy” (in the context of studies of personality and cognition). Reading further, I understood it might be

akin to what others call “docility”. Like me, maybe other readers are not familiar with this term, so please provide a short definition.

L135: since there are exceptions to most aspects of the protocols described here, I suggest moving the relevant information to the corresponding section regarding each test and leaving here at the beginning only the descriptions that apply to all test procedures. It would be interesting to know for each test, whether humans were in the testing room or whether the experiment was observed via camera.

L140-142: could you please elaborate a little more on how this protocol helped minimize the isolation stress? Was the bird aware of the cage-mate through sound or smell? I think this aspect is very important and I applaud the authors for thinking of this and implementing it. Explaining how this worked will help future readers to incorporate this concept (if not the exact protocols) in their studies.

L146: based on variation in body mass, metabolism and amount of food ingested prior to the fasting, I argue that food-depriving individuals prior to foraging tasks will have certainly increased motivation, but not standardised it. I suggest replacing “standardised” with “increased” or any similar concept.

L148: a dominance test usually involves some kind of experimental manipulations of dyadic encounters, whereas here it seems to have been an observation of interactions in a multi-bird cage with food-deprived birds. To avoid confusion and misinterpretation of what you actually did (which I think is perfectly fine), I suggest referring to this as “assessment of dominance hierarchies”.

Line 153-159: (dominance test) Is latency time to approach the feeder for the first time for each individual not measured?

L163-170: were these parameters also measured in absence of any object? How can you tell that e.g. a longer latency to the feeder is due to the presence of the object otherwise (and not, for example, to differences in general activity)? Also, it might be discussed, whether the novel object was placed close enough to the feeder to prevent the animal from feeding; from the video it looks like it was not adjacent to the feeder but facing it.

L185-186 (struggle test): The reader will be wondering, how often the birds are handled in the daily laboratory routine or for different testing procedures. Are the experimenters known to the animals?

L192-193: please report which is the life expectancy of zebra finches and do not force the reader to search for the cited paper. This information helps the reader put into context your results in terms of long-term repeatability.

L223-230: In the SI, only videos 1-6 are present, so all comments on these tests are based on the written description and figures only.

L254: Did the different personality traits correlate with others within and among individuals? Was there a behavioural syndrome that would justify the addition of the cognitive aspects in the collection of behavioural variables recorded so far?

Also, I could not find clear mentions of whether the performance in the problem-solving tests was repeatable? This is one of the main recommendation by Griffin et al. (2015), which informed many methodological choices in this study. If the cognitive trait is not repeatable, what is the point of investigating its relation to repeatable personality traits?

Finally, please consider a broader way to statistically analyse correlations for many different tests, a multi-trait analysis or a variable reduction approach.

L265-267: please clarify how these aspects relate to your main questions and hypotheses. Also, did you control for any fixed factors, like for examples sex of the individual or test round or order of presentation? Considering that only a fraction of the animals solved the tasks, not controlling for these aspects puts reliability of results obtained with a relatively low sample size into question.

Table S2-3: please report all the value for the personality tests that were conducted. Restricting the presentation of the results to these few tests prevents a complete assessment of the results.

L296-297: do non-solvers also include animals that did not try to solve the tasks? Or did all birds participate?

L336-337: my understanding is that the main question of the paper concerns the relationship between some personality traits and problem solving performance. This is again a long discussion of methodological aspects, that does not really address the implications for the observed connection between personality and innovation. Conjecturing on why some of the parameters are not repeatable contributes very little to the discussion of the results in the context of the main questions this study addresses. If I misunderstood which question is prominent for the authors, a revision of the introduction that helps readers focus more clearly on the relevant results will be good.

L386-387: please provide a definition of cognitive style, as this is the first time the term is used and it is not clear why all personality traits should have been involved in all tasks for them to be part of a cognitive style (or perhaps you meant syndrome?)

L398-399: please discuss why you think this was the case. Stating that personality traits were weakly connected to problem solving in a task-dependent way is a repetition of the results and does not add to the current understanding of the relationship.

L411-412: The conclusion of the study is weak and not comprehensive to the reader. A conclusion might include why it is important to consider your work and multiple personality traits for future studies, and what is the biological meaning of the connection.

Minor comments

Table 1: I suggest an extra column filled with + or - signs (or alternative signs) that indicate e.g. your expecting increasing aggressiveness to predict increasing problem-solving ability.

L136: "We conducted one behavioural assessment per day" per animal I assume?

L173 (mirror test) "a small piece of millet" can be misunderstood as small grain, better might be: piece of foxtail millet or just millet grains or millet.

L340-341: something seems to be wrong in this sentence "A post-hoc nonparametric Wilcoxon test indicated a significantly lower time to catch on average in 2018 ($\bar{x} = 21.48$ s) compared to 2016 ($\bar{x} = 14.73$ s)" - shouldn't 14s be a lower time to catch than 21s?

L358-359: habituation to captivity? Weren't they born in captivity? Probably adjustment to new housing conditions is what is intended?

SI Document 1: those exact measures might be reported directly in the methods section.

Decision letter (RSOS-210413.R0)

Dear Dr Barrett

The Editors assigned to your paper RSOS-210413 "Links between personality traits and problem-solving performance in zebra finches (*Taeniopygia guttata*)" have made a decision based on their reading of the paper and any comments received from reviewers.

Regrettably, in view of the reports received, the manuscript has been rejected in its current form. However, a new manuscript may be submitted which takes into consideration these comments.

We invite you to respond to the comments supplied below and prepare a resubmission of your manuscript. Below the referees' and Editors' comments (where applicable) we provide additional requirements. We provide guidance below to help you prepare your revision.

Please note that resubmitting your manuscript does not guarantee eventual acceptance, and we do not generally allow multiple rounds of revision and resubmission, so we urge you to make every effort to fully address all of the comments at this stage. If deemed necessary by the Editors, your manuscript will be sent back to one or more of the original reviewers for assessment. If the original reviewers are not available, we may invite new reviewers.

Please resubmit your revised manuscript and required files (see below) no later than 25-Nov-2021. Note: the ScholarOne system will 'lock' if resubmission is attempted on or after this deadline. If you do not think you will be able to meet this deadline, please contact the editorial office immediately.

Please note article processing charges apply to papers accepted for publication in Royal Society Open Science (<https://royalsocietypublishing.org/rsos/charges>). Charges will also apply to papers transferred to the journal from other Royal Society Publishing journals, as well as papers submitted as part of our collaboration with the Royal Society of Chemistry (<https://royalsocietypublishing.org/rsos/chemistry>). Fee waivers are available but must be requested when you submit your manuscript (<https://royalsocietypublishing.org/rsos/waivers>).

Thank you for submitting your manuscript to Royal Society Open Science and we look forward to receiving your resubmission. If you have any questions at all, please do not hesitate to get in touch.

on behalf of Dr Kimberley Mathot (Associate Editor) and Kevin Padian (Subject Editor)
openscience@royalsociety.org

Associate Editor Comments to Author (Dr Kimberley Mathot):

Comments to the Author:

Dear Dr. Barrett,

Thank you for submitting your article for consideration at Royal Society Open Science. Your paper has now been reviewed by three experts in the field. Each of them agree that the study design and data set are solid. However, each of the referees also raised significant concerns about the appropriateness of the analyses that were conducted, specifically in reference to the stated objectives. Because I believe addressing the referee concerns will involve substantial work and may dramatically alter the conclusions drawn from the study, I am recommending the manuscript be rejected with the possibility to resubmit.

I felt that the referee reports were particularly detailed and constructive, and I hope you will consider their comments and revise accordingly. I believe that the manuscript will be much stronger if you are able to address their points. In particular, the misalignment between the concept of cognitive syndromes (the outcome of co-evolution of behavioural and cognitive traits) and your current analytical approach (causal effect of behavioural traits on problem-solving) should be addressed. Structural equation modelling may be a more appropriate framework, or multivariate analyses that estimate trait covariances without assuming causality.

In addition to the referee comments, I have a few further points I would like to see addressed in a revision.

1) estimation of short versus long-term repeatability: there is no need to take averages from each time category. In fact, this can lead to substantial bias in repeatability estimates. Instead, you should consider using the method outlined by Araya-Ajoy et al. 2015. As you are not estimating individual reaction norms, it will require slight modification of the model parameterization from what is provided in that paper (e.g., including series, but not series | ID) as random effects.

Araya-Ajoy, Y. G., Mathot, K. J. and Dingemans, N. J. (2015). An approach to estimate short-term, long-term and reaction norm repeatability. *Methods in Ecology and Evolution* 6, 1462-1473.

2) Acknowledgement that your long-term repeatability involves both a long time interval (2 years), but also a change in context. My understanding is that birds were housed with mates only during the second set of measures (that is why you couldn't assess dominance over the long-term). This could result in lower long-term repeatability due to IxE (individual differences in reaction norms) in response to the changes in housing conditions rather than simply time. There isn't anything you can do to address this confound, but it should at least be acknowledged.

3) Please provide the rationale for key analytical decisions in the main text, not the ESM.

Reviewer comments to Author:

Reviewer: 1

Comments to the Author(s)

The present study assesses links between different personality traits and three different problem-solving tasks in captive zebra finches. Several personality traits predicted success in the problem-solving tasks but some associations were task-specific.

The manuscript is overall well written and addresses a relevant question that has recently received increasing attention; that is, whether personality traits and cognitive abilities are correlated. While the question is not new, the authors are among the firsts to assess such links across different problem-solving tasks.

I have several rather minor comments that I would like to see addressed.

L61-71: I would wish for an explanation as to why it would be important to understand the link between personality and cognition.

L73: Please change to 'problem'.

L83: It says 'typically' - which studies are the exception? Please mention those here.

L85: Please clarify which definition of personality is used. According to the abstract, personality differences are 'consistent differences across time or contexts...', suggesting they can be single traits; here it is suggested that they are 'multidimensional collection of behavioural traits'. Would be 'syndromes' more appropriated here?

L91 onwards: Several studies have already assessed personality traits across longer periods in zebra finches; these should be acknowledged here (e.g. 7 months: David et al. 2012; 3 months: Schuett et al. 2011; 1 year: Wuerz et al. 2015).

L131: I suggest using degree Celsius instead of Fahrenheit.

L 155: Could you give a more complete list of physical interactions that were used, please?

L193: Were birds tested in the same groups across the years?

L223: It seems I do not have access to video 7-9.

L259: Are the patterns / repeatability values similar if only the data of the first trial 2016 and 2018 are used? As repeatabilities in the short term are not that high, it seems more appropriate not to use the means.

Analyses/Results: Potential sex-differences in patterns have not been assessed, even though some studies have shown that repeatabilities of personality traits significantly differ between males

and females and that links between personality and cognitive abilities can be sex-specific (for latter, see also intro of current study). Even though the sample size is not huge, I suggest considering such potential sex differences in the analyses.

L265: Were model assumptions checked and met? Usually, latencies do not follow Gaussian error structures. Please clarify in the text.

Line 298-304: It is unclear if those are the maximal models or reduced models; if the first, it seems more appropriate to add this info to the stats section.

L304-313: I suggest adding the direction of effect, especially for those results that are not summarised in Fig. 3 captions.

Discussion: the section on repeatability of behaviour is quite long and could be shortened considerably.

SI Document 1: I suggest adding this information to Fig 2 and deleting this document here.

References:

David, M., Auclair, Y., & Cezilly, F. (2012). Assessing Short- and Long-Term Repeatability and Stability of Personality in Captive Zebra Finches Using Longitudinal Data. *Ethology*, 118(10), 932-942. doi: 10.1111/j.1439-0310.2012.02085.x

Schuett, W., Dall, S. R. X., & Royle, N. J. (2011). Pairs of zebra finches with similar 'personalities' make better parents. *Animal Behaviour*, 81(3), 609-618. doi: 10.1016/j.anbehav.2010.12.006

Wuerz, Y., & Krüger, O. (2015). Personality over ontogeny in zebra finches: long-term repeatable traits but unstable behavioural syndromes. *Frontiers in Zoology*, 12(1), 1-14.

Reviewer: 2

Comments to the Author(s)

Please see all comments in the attached file

Reviewer: 3

Comments to the Author(s)

This study examines 1) the long-term repeatability of personality traits, and 2) whether multiple cognitive tests correlate with personality measures. A battery of personality tests (dominance test, general activity measurement, novel object test, startle test, struggle test, mirror-interaction test) and a battery of problem-solving tests (string-pull test, wire-removal test, and a lid-flip test) were conducted on 41 captive zebra finches. Personality tests were repeated after a short interval, and partly repeated after two years. Results showed that most variables measured in the personality tests were consistent over a two years period. The success in problem-solving tests was best predicted by dominance and boldness, with less dominant but bolder birds more likely to solve a higher number of tests.

I think this is a very nice and thorough study, with the potential to advance scientific knowledge in the field. However the presentation of this work will need extensive revisions. Unfortunately, I think that the manuscript, data analyses and result interpretation all need further work before being ready for publication. The overall aim of the study does not become apparent yet, the take away message remains unclear and diffuse, the story still needs shaping, and conclusions seem at least disconnected from the results. In short, I do not think the authors do justice to the impressive amount of effort that went into this data collection.

General comments

From the way it is currently presented, it is unclear whether the authors consider this a methodological study, highlighting potential and difficulties of measuring multiple personality traits in relation to cognition, or actually a study investigating this connection and its meaning for cognitive and behavioural evolution and research. The two levels are constantly mixed. If this is a study whose results are intended to contribute to the understanding of individual variation in cognition (and I think it definitely has the potential to do this), there is no clear reasoning on why it was attempted, and what the obtained results mean in a biological sense. The results illustrate that there is a connection between some personality measures and problem solving, but why is

the presence of this connection considered proof that running multiple tests is important? I do agree that this is what results show, and also that this methodological approach is important, but I do not see an evident connection in how this is presented here. It would be good if the authors clarified what they think is the main point of this paper, a validation of a methodological approach or the biological relevance of this link. This will help to focus more clearly on the aspects they want to highlight, and to create a hierarchical order of importance of their findings in the narrative that will be clear to readers as well.

My second issue concerns the claim that multiple cognitive traits were assessed in combination with multiple measures of personality. The problem-solving tests were repeated (great!), and even though it is not clear if a similar assessment was carried out in 2018, this is a very important aspect that has all my approval. However, while the personality tests indeed address multiple aspects of animal personality, the three cognitive tasks are variations of the same test, a removal of something to access the food reward visible through a transparent barrier. The three tasks were targeting the same cognitive ability, and relied on similar cognitive mechanisms (L234). The variation was in the non-cognitive aspects such as motor actions required to solve them. The papers that the authors cite (e.g. Griffin et al., 2015) in support of this approach, actually recommend testing multiple cognitive traits along with their repeatability and other personality traits. While the authors often write “three problem-solving tests”, mentions of repeated tests in the context of the need for multiple cognitive assessments are misleading. My recommendation is that the authors clarify that the multiple traits measured here only refer to personality (to avoid overselling) and, more importantly, that they explain why they think is important multiple measures of personality in connection with a single, albeit repeated, cognitive task. At the moment the reader is left with the impression that the main reason is that some measures of personality correlate with the problem-solving performance and others are not, without going any deeper into the implications of this connection.

Another aspect that in my opinion requires revision is that, currently, the rationale for this study is barely explained. The authors cite the meta-analysis that found that while personality and cognition are related, the direction of this relationship is not the same across species, or tasks etc. From the way this is presented, I wonder if we should expect any kind of biological relationship to be so universal and unidirectional, and why the species- or task-dependence is considered problematic. Personally, I think it is great that further studies are conducted on the topic – I just could not understand what are the authors ideas on this, though, as they are not presented in the text beyond the simple statements that the relationship not always goes in the same direction and that, for this reason (?), multiple (personality) traits should be considered. This intellectual contribution in contextualising findings is, in my opinion, quite as necessary as robust results themselves.

Finally, I think that the thoroughness with which the personality tests were conducted should extend to the analysis of the data derived from them. The various tests yielded a wealth of behavioural observations on several variables – but of course not every single variable measured in, for example, a neophobia test is an indicator of neophobia, and most likely expresses a combination of various traits. Currently, all variables are presented as carrying the same significance and being indicators of only the personality trait after which the corresponding test is named. However, the study of animal personality requires a more critical approach towards the measured variables, and a careful interpretation and evaluation of the observed responses. Have these tests been validated in an ecological setting before concluding that they actually measure boldness, aggressiveness, etc.? If not, what does it tell us that an individual approaching a mirror has a better chance of solving a foraging task? One test can simultaneously be influenced by and thus measure two or more personality traits. Although a test that directly measures a targeted trait may be a desirable goal, in reality a test will likely be influenced by multiple traits at the same time (Carter et al., 2013; Réale et al., 2007). A critical re-evaluation of the variable presented as personality traits is required before this work could be considered ready for publication. On this, I recommend the review by Carter et al. (2013, *Biol. Rev.* (2013), 88, pp. 465–475. 465, doi: 10.1111/brv.12007) as a very valuable starting point to start the revision process.

Specific comments

L36-39: no connection between the first two sentences of the abstract. Why is it important to examine personality and cognition in the same study? I think this info should come before the methodological approaches used in the past, to place this study in the context of current knowledge.

L45-49: From this conclusion it seems that the aim of the study is to support the recommendation of conducting multiple assessments of personality, but it does not address why. Why is it important for the study of cognition to have robust proof that personality is linked to cognition? And why is it important to know under which circumstances and how specific personality traits affect performance in cognitive tests?

L60-116: In its current form, the introduction is a mixture of unconnected statements about personality being important for fitness, having a non-linear relationship with cognition, and methodological approaches. Considering that personality is the consistent among-individual variation in behavioural responses, I would have expected some mentions on why variation in cognitive performance is important, why we should be concerned with it, and why should we consider the other behavioural aspect of the individuals under study to properly assess the behavioural performance we use to infer cognitive processes and abilities. Instead there are a lot of disconnected sentences that do not reach the heart of the problem, in my opinion, but keep mixing the methodological level with the biological significance of the topics addressed. Of course, this does not detract from the merits of the study, but makes it much harder to appreciate them. I recommend an extensive revision of this section.

L61-71: the introductory paragraph does not seem to lead clearly to the research question. Why is this link considered relevant? And what were the basis to expect that the relationship between and cognition should be a universal constant, where most behavioural and cognitive aspects are indeed sex-, condition-, age-, motivation-, and context-dependent (L68-70)?

L70-71: please explain briefly the choice of this particular aspect of cognition, also in reference to the ongoing debate on whether problem-solving and innovation are indicators of cognitive ability (e.g. Chen et al., 2020; Camacho-Alpízar et al., 2020). While I personally support this view, I believe that in this context the debate should at least be acknowledged.

L73-74: please explain in what way innovation is crucial to find new resources, it does not transpire clearly from this sentence (“Innovation, the ability to solve novel problems or to find new solutions to an old problems [8], is crucial for locating and using novel resources in a changing environment”) and is not mentioned elsewhere in the intro.

L72-77: these three sentences are a good example of the issues I find with the presentation of this study. The frequent oscillation between biological relevance of the topic addressed here – the link between personality and innovation, which is often mentioned but never addressed in depth – and methodological aspects. I believe that if the authors find the connection they investigate meaningful in a biological sense, it should be addressed beyond the mention of the one meta-analyses that found that the direction of the relationship varies. And the significance of this relation should be kept separate from the methodological considerations.

L76-77: please provide support for these statements, as they refer to existing knowledge and previous studies.

L77-83: after describing the variation in results, I suggest making it explicit why this is considered problematic. Why would it be good to have a one-way expectation for the link between personality and problem-solving performance? And do the authors think that given the diversity of morphology, environments, social structure in the listed taxa, such a one-way relationship is really possible?

L89-90: I disagree: personality might affect the performance, the way in which a cognitive task is approached and solved, not the ability itself.

L91-93: And what is the significance of having some personality measures repeatable after an interval of 2 years? What does it mean for zebra finches personality, its evolution, its trade-off with plasticity? And what methodological conclusions can be derived from knowing that zebra

finches maintain some of their behavioural responses consistent throughout a significant part of their life?

L101: the fact that a study has not been conducted in a certain species is not a sufficient reason to justify one. Please provide reasons why you consider this species a suitable model to address your question, and why it should provide answers that are valid across taxa, as it seems implied in the introduction and abstract in their current form.

L104-107: "Individuals capable of accessing food resources (such as more dominant individuals), especially when food is ephemeral, may possess an advantage over other individuals to expand the amount or type of food available to them. We filled this gap in knowledge by [...]" I see no gap in knowledge in the previous sentence, and the statement does not really explain the study, especially since foraging or fitness consequences are not assessed.

L112-113: Why is it important to have multiple measures of both cognitive performance and personality? What does that add to both fields? A reader should not be forced to Griffin et al. (2015) to start to have answers on this. Please revise the introduction so that the relevance of the study is clear, even to reader who are not familiar with the studies you cite.

Table 1: I am unclear how some of these predictions were developed. If the relationship between personality and innovation varies across species and no previous study was conducted in zebra finches, how did you choose which studies to rely on to inform the predictions (e.g. on dominance?).

There is a single prediction clumping together activity, neophobia and aggressiveness; are these traits considered interchangeable? Given the ecological meaning of each, and how very distinct form one another they are, I found it difficult to understand why they are considered together in here – unless this was done in light of the obtained results? If that is the case, please revise, as predictions should arise well before knowing the study outcome.

The predictions on risk-taking are mixing the methodological approach (how this trait is measured – in a fraction of studies) with its significance and potential to affect problem-solving performance and innovation propensity. Lastly, I have never come across the term "obstinacy" (in the context of studies of personality and cognition). Reading further, I understood it might be akin to what others call "docility". Like me, maybe other readers are not familiar with this term, so please provide a short definition.

L135: since there are exceptions to most aspects of the protocols described here, I suggest moving the relevant information to the corresponding section regarding each test and leaving here at the beginning only the descriptions that apply to all test procedures. It would be interesting to know for each test, whether humans were in the testing room or whether the experiment was observed via camera.

L140-142: could you please elaborate a little more on how this protocol helped minimize the isolation stress? Was the bird aware of the cage-mate through sound or smell? I think this aspect is very important and I applaud the authors for thinking of this and implementing it. Explaining how this worked will help future readers to incorporate this concept (if not the exact protocols) in their studies.

L146: based on variation in body mass, metabolism and amount of food ingested prior to the fasting, I argue that food-depriving individuals prior to foraging tasks will have certainly increased motivation, but not standardised it. I suggest replacing "standardised" with "increased" or any similar concept.

L148: a dominance test usually involves some kind of experimental manipulations of dyadic encounters, whereas here it seems to have been an observation of interactions in a multi-bird cage with food-deprived birds. To avoid confusion and misinterpretation of what you actually did (which I think is perfectly fine), I suggest referring to this as "assessment of dominance hierarchies".

Line 153-159: (dominance test) Is latency time to approach the feeder for the first time for each individual not measured?

L163-170: were these parameters also measured in absence of any object? How can you tell that e.g. a longer latency to the feeder is due to the presence of the object otherwise (and not, for

example, to differences in general activity)? Also, it might be discussed, whether the novel object was placed close enough to the feeder to prevent the animal from feeding; from the video it looks like it was not adjacent to the feeder but facing it.

L185-186 (struggle test): The reader will be wondering, how often the birds are handled in the daily laboratory routine or for different testing procedures. Are the experimenters known to the animals?

L192-193: please report which is the life expectancy of zebra finches and do not force the reader to search for the cited paper. This information helps the reader put into context your results in terms of long-term repeatability.

L223-230: In the SI, only videos 1-6 are present, so all comments on these tests are based on the written description and figures only.

L254: Did the different personality traits correlate with others within and among individuals?

Was there a behavioural syndrome that would justify the addition of the cognitive aspects in the collection of behavioural variables recorded so far?

Also, I could not find clear mentions of whether the performance in the problem-solving tests was repeatable? This is one of the main recommendation by Griffin et al. (2015), which informed many methodological choices in this study. If the cognitive trait is not repeatable, what is the point of investigating its relation to repeatable personality traits?

Finally, please consider a broader way to statistically analyse correlations for many different tests, a multi-trait analysis or a variable reduction approach.

L265-267: please clarify how these aspects relate to your main questions and hypotheses. Also, did you control for any fixed factors, like for examples sex of the individual or test round or order of presentation? Considering that only a fraction of the animals solved the tasks, not controlling for these aspects puts reliability of results obtained with a relatively low sample size into question.

Table S2-3: please report all the value for the personality tests that were conducted. Restricting the presentation of the results to these few tests prevents a complete assessment of the results.

L296-297: do non-solvers also include animals that did not try to solve the tasks? Or did all birds participate?

L336-337: my understanding is that the main question of the paper concerns the relationship between some personality traits and problem solving performance. This is again a long discussion of methodological aspects, that does not really address the implications for the observed connection between personality and innovation. Conjecturing on why some of the parameters are not repeatable contributes very little to the discussion of the results in the context of the main questions this study addresses. If I misunderstood which question is prominent for the authors, a revision of the introduction that helps readers focus more clearly on the relevant results will be good.

L386-387: please provide a definition of cognitive style, as this is the first time the term is used and it is not clear why all personality traits should have been involved in all tasks for them to be part of a cognitive style (or perhaps you meant syndrome?)

L398-399: please discuss why you think this was the case. Stating that personality traits were weakly connected to problem solving in a task-dependent way is a repetition of the results and does not add to the current understanding of the relationship.

L411-412: The conclusion of the study is weak and not comprehensive to the reader. A conclusion might include why it is important to consider your work and multiple personality traits for future studies, and what is the biological meaning of the connection.

Minor comments

Table 1: I suggest an extra column filled with + or – signs (or alternative signs) that indicate e.g. your expecting increasing aggressiveness to predict increasing problem-solving ability.

L136: “We conducted one behavioural assessment per day” per animal I assume?

L173 (mirror test) “a small piece of millet” can be misunderstood as small grain, better might be: piece of foxtail millet or just millet grains or millet.

L340-341: something seems to be wrong in this sentence “A post-hoc nonparametric Wilcoxon test indicated a significantly lower time to catch on average in 2018 (\bar{x} = 21.48 s) compared to 2016 (\bar{x} = 14.73 s)” – shouldn't 14s be a lower time to catch than 21s?

L358-359: habituation to captivity? Weren't they born in captivity? Probably adjustment to new housing conditions is what is intended?

SI Document 1: those exact measures might be reported directly in the methods section.

===PREPARING YOUR MANUSCRIPT===

===PREPARING YOUR REVISION IN SCHOLARONE===

Author's Response to Decision Letter for (RSOS-210413.R0)

See Appendix B.

RSOS-212001.R0

Review form: Reviewer 2

Is the manuscript scientifically sound in its present form?

Yes

Are the interpretations and conclusions justified by the results?

No

Is the language acceptable?

Yes

Do you have any ethical concerns with this paper?

Yes

Have you any concerns about statistical analyses in this paper?

No

Recommendation?

Major revision is needed (please make suggestions in comments)

Comments to the Author(s)

In general, the authors carefully attended to my previous comments and the current version of the manuscript has improved a lot. Especially the introduction now makes the study goals clear and the methods are much clearer now. I am left with one major concern and a handful of smaller suggestions to improve readability.

All lines refer to the manuscript with highlighted changes and to the original line numbers included in the word or similar program.

My major concern is the analysis of the cognitive tests and their association with personality traits. At the moment, the reader is left to wonder whether the three cognitive tests really all reflect problem-solving behaviour or describe various different cognitive traits in zebra finches. I suggest to investigate this in more detail (see my comment to line 422). This knowledge should then inform how associations with the different personality measures are analysed. At the moment the probability to solve is analysed separately for all three tests while the average latency to solve across all three tests is considered in 1 model. If all three tests indeed reflect problem-solving behaviour, then one model including all three tests should investigate personality associations for solving probability and a second model should investigate personality and solving latency. In addition, I strongly suggest to use a mixed model framework then instead of averaging across several test situations. Otherwise, it will not be possible to separate within- from between individual effects (see for a similar discussion on animal personality and behavioural syndromes for example Brommer (2013)). If, however, the three tests measure different aspects of cognition, the analyses should be split up for the tests consistently, i.e. the model explaining average latency should not be used. Based on these analyses, I would then also advice to re-write the second part of the discussion starting in line 527. At the moment, this part of the introduction still leaves the reader wondering what we have learnt about the association of personality and cognition/ problem-solving in particular.

Brommer, J. E. (2013). On between-individual and residual (co) variances in the study of animal personality: are you willing to take the “individual gambit”? *Behavioral Ecology and Sociobiology*, 67(6), 1027-1032.

- Line 36-37: this sentence does not make sense here since cognitive abilities have not yet been introduced. Move to line 41
- Line 41: measuring instead of measured
- Check consistency of referencing, e.g. line 84, change in style
- Line 113: add “,” after: Thus
- Line 147-148: personality does not fall under animal cognition, please rephrase sentence

- Line 193-194: I cannot judge if permits would needed to have been acquired under the local laws. In the European Union, experiments such as performed here would need a formal application to the responsible authorities

- Line 246-247: is there something missing from this sentence? I don't understand the meaning. In addition, the sentence comparing response between trials is copy-pasted in the results section. I think it makes more sense there and should be deleted here.

- Line 353: there is a verb missing in this sentence

- Line 368-369: what kind of correlation was used here? And how were repeated measures from individuals dealt with?

- Line 399: why is there a citation in the results section? I think the results reported in this sentence were collected in this study? In addition, I do not understand why a separate Wilcoxon test was performed here. The result whether or not trial performance differed from each other should be available in the model estimating the repeatability since “trial” was included as fixed effect. Or was “day 1-6” included in the model here instead of trial (compare description lines 242-243)? In addition, I'm wondering, to estimate the repeatability here, were both daily trials (i.e. without object and with object) included in the analysis? This might explain a non-significant repeatability since these are quite different situations.

- Lines 412-420: I suggest to delete this section since just comparing confidence intervals like this is not sufficient to determine if the sexes differ in repeatability or not. As the investigation of potential sex-differences was not a main aim of the study, losing this paragraph will not affect the general conclusions.

- Line 422: I suggest to additionally test for repeatability after excluding all individuals that never approached the test setup. Currently, non-solvers could include animals that lack motivation to interact with the test setup as well as animals who are really just bad innovators and therefore score the maximum latency. In addition, I suggest to run a model checking the probability to solve, i.e. assuming a binomial distribution to inform us more about the generality of the tests used here to investigate problem-solving behaviour.

- Line 430: please test for variance inflation factors to exclude the possibility that covariation between personality traits of different tests influence the model outcome.

- Line 522-524: I suggest to delete this sentence since the corresponding statistical analysis is not appropriate to test sex-differences in repeatability (see also my comment to lines 412-420).

Review form: Reviewer 3

Is the manuscript scientifically sound in its present form?

No

Are the interpretations and conclusions justified by the results?

No

Is the language acceptable?

Yes

Do you have any ethical concerns with this paper?

No

Have you any concerns about statistical analyses in this paper?

Yes

Recommendation?

Major revision is needed (please make suggestions in comments)

Comments to the Author(s)

In RSOS-212001 the authors revised their previous manuscript describing a study on long-term repeatability of personality traits, and their link to innovative problem solving. The revisions to the text have, in my opinion, improved the readability of the text, clarified the aims and research questions, and provided an overall better structure for the manuscript. Comments from the previous round of review have been attended, in terms of additional information and clarifications. Methods are now well described and easy to follow.

However, I am afraid that when it comes to the most substantial points raised in the previous round of review, the current revision is not complete or appropriate, making it very difficult to re-evaluate the merit of this contribution. I think that the alternative analytical approaches suggested would have indeed implied an advancement in the way these type of data are used to answer timely questions. The approach the authors chose might have been used in the past, but it does not mean it is the most suited for such a complex dataset as this.

Additionally, my concern about the multiplicity of the cognitive tasks was not addressed in a convincing manner, in my opinion. I did (and still do) think that the three cognitive tasks used here represent variations of the same test, a removal of something to access the food reward visible through a transparent barrier. The three tasks were targeting the same cognitive ability, and relied on similar cognitive mechanisms, as the authors stated in the first draft. The variation was in the non-cognitive aspects such as motor actions required to solve them. Now the authors state that slightly different cognitive aspects might have been involved in the three tasks, and that is why they were presented. They list the fact that sometimes dominance and sometimes aggressiveness were weakly correlated with success, in support of different cognitive aspects being required in each task (L574-577). I consider this approach quite problematic. First of all, if it is not known which aspects of innovative problem solving the three tasks address, this aspect should likely be illuminated before using such tasks to investigate the connection between problem solving and non-cognitive aspects like personality. Secondly, I do not see how non-cognitive aspects (be it motor diversity or personality) involved in the performance might constitute support for the three tasks being different from a cognitive perspective. In sum, I still think calling these variants "multiple cognitive tasks" misleading, both in describing the width of the study and especially in the results interpretation and recommendation for the future. That it is a repeated test, like the behavioural ones, is agreed. But that these are diverse, multiple cognitive tasks addressing innovation in different ways, is not.

Finally, while the general readability has increased, the question of why and how personality should be linked to cognitive performance is still dealt with quite superficially, and does not go beyond the citation of previous studies stating that such links exist and are to be expected.

Specific comments

L85-87: There is still no explicit explanation of how personality may correlate with innovation, except for mentions of the review. I think it is important to illustrate in detail which aspects of personality might co-vary with cognitive performance, and through which mechanisms. Listing examples (e.g. L91-97) is not quite the same as providing an explanation for the patterns.

L107-109: This clarifies the study aims better. But there is still no biological reason for wanting to know more about long-term repeatability of behavioural responses.

L151-155: In the previous sentence the authors describe how dominant individuals that arrive at the food first, have more potential to innovate. But the "necessity drives innovation" hypothesis describes exactly the opposite scenario, with less competitive animals having to resort to innovation to compensate for restricted access to common food sources. In the rest of the text the necessity-drives-innovation hypothesis is correctly referred to, so it might have been a temporary mix-up here.

Additionally, dominance is not a synonym of aggressiveness in many species - but could be that in zebra finches more aggressive animals (and not e.g. older or bigger) are at the top of the hierarchy. If this is the case here, it should be stated, or the logical progression of arguments is hard to follow.

L266-270: Mentions of the ecological relevance if this test appear the very last time this is mentioned in the discussion (L563-566). Until then it is not clear at all what is the supposed ecological relevance of the behaviours measured in this test. I am not inferring that there is none, just pointing out that this information is missing and it gets in the way of understanding the significance of the results.

Further, in my previous round of comments I asked about the existence of studies validating these tests in an ecological context. In order to talk about coevolution with cognition and selective pressure on certain personality traits, we need to know whether the behaviours measured in the lab indeed translate into real-world responses to environmental stimuli. Do these tests predict how zebra finches respond to novelty, startling events, etc. in the wild?

L412: the other part of Rew. 1 suggestion, concerning links between personality and cognitive abilities being sometimes sex-specific, was not addressed either in the ms or in the response.

Tables 3-4: if I understood correctly, repeatabilities reported here are adjusted for Trial and (where possible) Year. So it would be good to state that these values are for adjusted repeatabilities.

L478-480: One problem the field of animal personality is facing, is indeed the lack of consistent terminology. What here (and indeed in a few other papers) is called obstinacy, i.e. the response to human capture or handling, is more often described as "docility" and has received more attention than here implied (e.g. Dingemanse and Réale, 2005; Petelle et al., 2013, 2015; Hall et al, 2015; Guenther et al., 2018).

L489-491: I do not think this is what the results show. Most variables of each test were repeatable.

L563-566: This arrives quite late (see my comment above). Additionally, stating that only another study on something exists is quite risky (e.g. see Edwards et al. 2017, for a study on repeatability and heritability of obstinacy in warblers).

L639 (very minor): I believe a comment escaped a final edit of the text.

Decision letter (RSOS-212001.R0)

Dear Dr Barrett

The Editors assigned to your paper RSOS-212001 "Links between personality traits and problem-solving performance in zebra finches (*Taeniopygia guttata*)" have now received comments from reviewers and would like you to revise the paper in accordance with the reviewer comments and any comments from the Editors. Please note this decision does not guarantee eventual acceptance.

Please submit your revised manuscript and required files (see below) no later than 21 days from today's (ie 02-Feb-2022) date. Note: the ScholarOne system will 'lock' if submission of the revision is attempted 21 or more days after the deadline. If you do not think you will be able to meet this deadline please contact the editorial office immediately.

on behalf of Dr Kimberley Mathot (Associate Editor) and Kevin Padian (Subject Editor)
openscience@royalsociety.org

Editor comments:

Thank you for your resubmission, and for your diligence in addressing the comments of the reviewers. As you see from the AE's comments there are still some issues to resolve that I hope you can take care of easily. If you need more time than the allotted window, please contact the editorial office. I remind you that your next version needs to satisfy all concerns of the reviewers and AE, or we will be unable to consider it further. Thanks and best wishes.

Associate Editor Comments to Author (Dr Kimberley Mathot):

Associate Editor

Comments to the Author:

Thank you for submitting a revised version of your manuscript for consideration at RSOS. The revised manuscript has now been reviewed by two of the referees who commented on the original submission. Both referees feel that the current version of the manuscript is much improved. Specifically, the aims of the study are laid out more clearly in the introduction, and the methods are described in better detail. Both referees still have a number of points that I would like to see addressed before the manuscript can be accepted. Most of these are minor. However, one of the more significant points I would like to see addressed is the concern raised about whether (or the extent to which) the three cognitive tasks are independent measures of cognitive ability. There are various ways that you could address this. In fact, part of your results seem to address this, but it's not entirely clear to me the models from which these results were derived (lines 427-429). I would suggest a two-phase approach to analyze the non-independence of the three cognitive tasks.

1) is the probability of solving repeatable

$\text{glmer}(\text{solved} \sim \text{task_type} + (1 | \text{ID}), \text{family} = \text{binomial})$, where solved is (1 = yes, 0 = no), and task type is "string pull", "wire removal" or "lid flip".

From this model you can assess the repeatability of solving yes/no as a way of inferring the extent to which the tasks may be independent.

2) for birds that solved, is the time to solve repeatable.

$\text{glmer}(\text{time to solve} \sim \text{task_type} + (1 | \text{ID}), \text{family} = \text{poisson})$.

Note, you may assign max time values for non-solving birds. You may also find that transforming the data and then modelling gaussian errors works better if the poisson models are overdispersed.

Note that any degree of repeatability will give you information about the potential non-independent between these different test types. Also- it is important that other predictors are not included in these models (e.g., behaviours, sex), as that will erode part of the among-individual variation. Regardless of the finding, it would be worth explicitly acknowledging the degree of (non-)independence in the results and discussion.

Reviewer #1 also points out that the analytical decision to use means for behavioural traits rather than the repeated measures comes with several limitations. They are absolutely correct. However, given your data structure, the analytical approach that would allow you to use repeated measures of behaviour with a single measure per cognitive task would require a multivariate model, and your sampling design (number of individuals, number of repeats per individual) is not sufficient for you to have the power to do this. Instead, I would ask that you address this point by acknowledging the consequences that using a single measure per individual can have for parameter estimates. Referee #1 has provided you with some relevant references to address this.

In addition to the comments provided by the referees, I have the following comments I would like you to address in the revised version:

1. Please refrain from referring to specific behavioural traits as "personality traits". Personality is a statistical property of a population (i.e., repeatable among-individual variation) – and any behaviour for which such repeatable among-individual variation exists can be studied in an animal personality framework. Further – studying a trait that is commonly studied in animal personality (e.g., boldness, neophobia) cannot be assumed a priori to harbour significant among-individual variation. You assess the repeatability of the traits you studied – so it won't change any of your conclusions, but the terminology should be corrected.

For example, on line 166 you write, "are personality traits repeatable not only over the short-term but also over the relatively longer-term?". By definition, the answer must be yes otherwise the

traits cannot be studied in an animal personality framework. Instead, you can write “1) we estimated both the short- and long-term repeatability of X, Y, Z, and asked whether repeatable variation in X, Y, Z as associated with differences in problem-solving performance.

2. For Poisson models (Line 381)- please confirm that you checked for overdispersion.

Reviewer comments to Author:

Reviewer: 2

Comments to the Author(s)

In general, the authors carefully attended to my previous comments and the current version of the manuscript has improved a lot. Especially the introduction now makes the study goals clear and the methods are much clearer now. I am left with one major concern and a handful of smaller suggestions to improve readability.

All lines refer to the manuscript with highlighted changes and to the original line numbers included in the word or similar program.

My major concern is the analysis of the cognitive tests and their association with personality traits. At the moment, the reader is left to wonder whether the three cognitive tests really all reflect problem-solving behaviour or describe various different cognitive traits in zebra finches. I suggest to investigate this in more detail (see my comment to line 422). This knowledge should then inform how associations with the different personality measures are analysed. At the moment the probability to solve is analysed separately for all three tests while the average latency to solve across all three tests is considered in 1 model. If all three tests indeed reflect problem-solving behaviour, then one model including all three tests should investigate personality associations for solving probability and a second model should investigate personality and solving latency. In addition, I strongly suggest to use a mixed model framework then instead of averaging across several test situations. Otherwise, it will not be possible to separate within- from between individual effects (see for a similar discussion on animal personality and behavioural syndromes for example Brommer (2013)). If, however, the three tests measure different aspects of cognition, the analyses should be split up for the tests consistently, i.e. the model explaining average latency should not be used. Based on these analyses, I would then also advice to re-write the second part of the discussion starting in line 527. At the moment, this part of the introduction still leaves the reader wondering what we have learnt about the association of personality and cognition/ problem-solving in particular.

Brommer, J. E. (2013). On between-individual and residual (co) variances in the study of animal personality: are you willing to take the “individual gambit”?. *Behavioral Ecology and Sociobiology*, 67(6), 1027-1032.

- Line 36-37: this sentence does not make sense here since cognitive abilities have not yet been introduced. Move to line 41

- Line 41: measuring instead of measured

- Check consistency of referencing, e.g. line 84, change in style

- Line 113: add “,” after: Thus

- Line 147-148: personality does not fall under animal cognition, please rephrase sentence

- Line 193-194: I cannot judge if permits would needed to have been acquired under the local laws. In the European Union, experiments such as performed here would need a formal application to the responsible authorities

- Line 246-247: is there something missing from this sentence? I don't understand the meaning. In addition, the sentence comparing response between trials is copy-pasted in the results section. I think it makes more sense there and should be deleted here.
- Line 353: there is a verb missing in this sentence
- Line 368-369: what kind of correlation was used here? And how were repeated measures from individuals dealt with?
- Line 399: why is there a citation in the results section? I think the results reported in this sentence were collected in this study? In addition, I do not understand why a separate Wilcoxon test was performed here. The result whether or not trial performance differed from each other should be available in the model estimating the repeatability since "trial" was included as fixed effect. Or was "day 1-6" included in the model here instead of trial (compare description lines 242-243)? In addition, I'm wondering, to estimate the repeatability here, were both daily trials (i.e. without object and with object) included in the analysis? This might explain a non-significant repeatability since these are quite different situations.
- Lines 412-420: I suggest to delete this section since just comparing confidence intervals like this is not sufficient to determine if the sexes differ in repeatability or not. As the investigation of potential sex-differences was not a main aim of the study, losing this paragraph will not affect the general conclusions.
- Line 422: I suggest to additionally test for repeatability after excluding all individuals that never approached the test setup. Currently, non-solvers could include animals that lack motivation to interact with the test setup as well as animals who are really just bad innovators and therefore score the maximum latency. In addition, I suggest to run a model checking the probability to solve, i.e. assuming a binomial distribution to inform us more about the generality of the tests used here to investigate problem-solving behaviour.
- Line 430: please test for variance inflation factors to exclude the possibility that covariation between personality traits of different tests influence the model outcome.
- Line 522-524: I suggest to delete this sentence since the corresponding statistical analysis is not appropriate to test sex-differences in repeatability (see also my comment to lines 412-420).

Reviewer: 3

Comments to the Author(s)

In RSOS-212001 the authors revised their previous manuscript describing a study on long-term repeatability of personality traits, and their link to innovative problem solving. The revisions to the text have, in my opinion, improved the readability of the text, clarified the aims and research questions, and provided an overall better structure for the manuscript. Comments from the previous round of review have been attended, in terms of additional information and clarifications. Methods are now well described and easy to follow.

However, I am afraid that when it comes to the most substantial points raised in the previous round of review, the current revision is not complete or appropriate, making it very difficult to re-evaluate the merit of this contribution. I think that the alternative analytical approaches suggested would have indeed implied an advancement in the way these type of data are used to answer timely questions. The approach the authors chose might have been used in the past, but it does not mean it is the most suited for such a complex dataset as this.

Additionally, my concern about the multiplicity of the cognitive tasks was not addressed in a convincing manner, in my opinion. I did (and still do) think that the three cognitive tasks used here represent variations of the same test, a removal of something to access the food reward visible through a transparent barrier. The three tasks were targeting the same cognitive ability, and relied on similar cognitive mechanisms, as the authors stated in the first draft. The variation was in the non-cognitive aspects such as motor actions required to solve them. Now the authors state that slightly different cognitive aspects might have been involved in the three tasks, and that is why they were presented. They list the fact that sometimes dominance and sometimes aggressiveness were weakly correlated with success, in support of different cognitive aspects being required in each task (L574-577). I consider this approach quite problematic. First of all, if it is not known which aspects of innovative problem solving the three tasks address, this aspect should likely be illuminated before using such tasks to investigate the connection between problem solving and non-cognitive aspects like personality. Secondly, I do not see how non-cognitive aspects (be it motor diversity or personality) involved in the performance might constitute support for the three tasks being different from a cognitive perspective. In sum, I still think calling these variants "multiple cognitive tasks" misleading, both in describing the width of the study and especially in the results interpretation and recommendation for the future. That it is a repeated test, like the behavioural ones, is agreed. But that these are diverse, multiple cognitive tasks addressing innovation in different ways, is not.

Finally, while the general readability has increased, the question of why and how personality should be linked to cognitive performance is still dealt with quite superficially, and does not go beyond the citation of previous studies stating that such links exist and are to be expected.

Specific comments

L85-87: There is still no explicit explanation of how personality may correlate with innovation, except for mentions of the review. I think it is important to illustrate in detail which aspects of personality might co-vary with cognitive performance, and through which mechanisms. Listing examples (e.g. L91-97) is not quite the same as providing an explanation for the patterns.

L107-109: This clarifies the study aims better. But there is still no biological reason for wanting to know more about long-term repeatability of behavioural responses.

L151-155: In the previous sentence the authors describe how dominant individuals that arrive at the food first, have more potential to innovate. But the "necessity drives innovation" hypothesis describes exactly the opposite scenario, with less competitive animals having to resort to innovation to compensate for restricted access to common food sources. In the rest of the text the necessity-drives-innovation hypothesis is correctly referred to, so it might have been a temporary mix-up here.

Additionally, dominance is not a synonym of aggressiveness in many species - but could be that in zebra finches more aggressive animals (and not e.g. older or bigger) are at the top of the hierarchy. If this is the case here, it should be stated, or the logical progression of arguments is hard to follow.

L266-270: Mentions of the ecological relevance if this test appear the very last time this is mentioned in the discussion (L563-566). Until then it is not clear at all what is the supposed ecological relevance of the behaviours measured in this test. I am not inferring that there is none, just pointing out that this information is missing and it gets in the way of understanding the significance of the results.

Further, in my previous round of comments I asked about the existence of studies validating these tests in an ecological context. In order to talk about coevolution with cognition and selective pressure on certain personality traits, we need to know whether the behaviours measured in the lab indeed translate into real-world responses to environmental stimuli. Do these tests predict how zebra finches respond to novelty, startling events, etc. in the wild?

L412: the other part of Rew. 1 suggestion, concerning links between personality and cognitive abilities being sometimes sex-specific, was not addressed either in the ms or in the response.

Tables 3-4: if I understood correctly, repeatabilities reported here are adjusted for Trial and (where possible) Year. So it would be good to state that these values are for adjusted repeatabilities.

L478-480: One problem the field of animal personality is facing, is indeed the lack of consistent terminology. What here (and indeed in a few other papers) is called obstinacy, i.e. the response to human capture or handling, is more often described as "docility" and has received more attention than here implied (e.g. Dingemans and Réale, 2005; Petelle et al., 2013, 2015; Hall et al, 2015; Guenther et al., 2018).

L489-491: I do not think this is what the results show. Most variables of each test were repeatable.

L563-566: This arrives quite late (see my comment above). Additionally, stating that only another study on something exists is quite risky (e.g. see Edwards et al. 2017, for a study on repeatability and heritability of obstinacy in warblers).

L639 (very minor): I believe a comment escaped a final edit of the text.

===PREPARING YOUR MANUSCRIPT===

If you have been asked to revise the written English in your submission as a condition of publication, you must do so, and you are expected to provide evidence that you have received language editing support. The journal would prefer that you use a professional language editing service and provide a certificate of editing, but a signed letter from a colleague who is a fluent speaker of English is acceptable. Note the journal has arranged a number of discounts for authors using professional language editing services (<https://royalsociety.org/journals/authors/benefits/language-editing/>).

===PREPARING YOUR REVISION IN SCHOLARONE===

To revise your manuscript, log into <https://mc.manuscriptcentral.com/rsos> and enter your Author Centre - this may be accessed by clicking on "Author" in the dark toolbar at the top of the

page (just below the journal name). You will find your manuscript listed under "Manuscripts with Decisions". Under "Actions", click on "Create a Revision".

Author's Response to Decision Letter for (RSOS-212001.R0)

See Appendix C.

Decision letter (RSOS-212001.R1)

Dear Dr Barrett,

It is a pleasure to accept your manuscript entitled "Links between personality traits and problem-solving performance in zebra finches (*Taeniopygia guttata*)" in its current form for publication in Royal Society Open Science. The comments from the Editors are included at the foot of this letter.

on behalf of Dr Kimberley Mathot (Associate Editor) and Kevin Padian (Subject Editor)
openscience@royalsociety.org

Associate Editor Comments to Author (Dr Kimberley Mathot):

Thank you for submitting your revised manuscript and for clearly detailing how you addressed each of the comments from myself and the two referees. I am satisfied that all the comments have been addressed adequately, and I have only a few very minor edits I would like you to implement:

Lines 419-420 (of the marked version of the MS) - please be consistent with the number of significant digits provided for the two repeatability estimates.

Lines 551-553: The sentence in its current form misses some nuance. I recommend this alternative phrasing: "This approach ignores the error variance around individual mean behaviour, and can lead to spurious significance". [Relevant references for this statement are: Houslay TM, Wilson AJ, 2017. Avoiding the misuse of BLUP in behavioural ecology. *Behav Ecol* 28:948-952. doi: 10.1093/beheco/ax023. and Dingemanse NJ, Wright J, 2020. Criteria for acceptable studies of animal personality and behavioural syndromes. *Ethology* 126:865-869. doi: 10.1111/eth.13082.]

Lines 592: correct number of anonymous referees was 3.

In Supplementary Table 1: one instance of "Hierarchies" is spelled incorrectly (missing the second "i").

Appendix A

Manuscript RSOS-210413 investigates associations between several personality traits such as dominance, activity or neophobia and problem-solving ability in zebra finches. In addition, the authors emphasize their estimates of long-term repeatability estimates. In general, the topic of the study is timely and will likely attract many readers. The number of animals tested and the design of experiment is sound and carefully chosen based on existing literature. Thus, I am convinced that the study offers a valuable dataset which will make a good contribution to the literature. In its current state however, I cannot recommend the manuscript for publication. Mainly, I have serious concerns about the statistical analysis carried out and therefore in the interpretation of results. In addition, methods are often described superficially which makes it hard to judge their appropriateness. Given the potential merit I see in the data, I would be happy to review a substantially revised version of the article.

General comments:

The authors place the study into the area of cognitive syndromes in the background but later test for the predictive effects of personality traits on cognitive traits. There are conceptual differences between these two approaches. The concept of a cognitive syndrome relies on correlations between personality and cognitive traits (for example because both might have been shaped by correlational selection or through pleiotropic effects). Thus, correlations (i.e. without a priori assumptions of predictive effects) should be used (and here, following the concept of behavioural syndromes, assumptions primarily apply to between-individual rather than phenotypic correlations). The statistical methods applied here on the other hand, assume that variation in personality traits predict, i.e. generate variation in cognitive traits. I suggest to apply a multivariate mixed-effect modelling approach throughout.

The statistical analyses are hard to follow and seem inappropriate in several places. Therefore, conclusions and discussion cannot be reliably reviewed at this stage.

Background:

- I am missing a part that explains why it is important to study the relationship between personality and cognition in a biological context. In addition, a short introduction on why these seemingly different traits should be associated is missing. "Individual variation in cognition [...] likely relates to personality types" is a bit short.
- To understand why specific personality measures were chosen to be tested here, it would be necessary to know that the scientific hypothesis behind the expected association between certain cognitive traits and certain personality, relies on a connection of both types of traits to a shared risk-reward trade-off. I can for example not understand why general activity should be measured since I fail to see the connection to a risk-reward trade-off. To prevent more studies reporting mixed results in the future, the clear connection to the scientific background should always be made crystal clear. Since we have well-developed scientific hypotheses that can be tested, we should be past the stage where more or less random trait associations can be reported.
- The connection between innovative problem-solving and cognition as a whole is not introduced.

Methods:

- Lines 121-128: The timeline is a bit unclear. The authors say testing started 4 months after obtaining birds from the breeder but a few lines further down, a two-week habituation period is mentioned. How do these relate to each other? In addition, it is not fully clear in which cages/ group sizes animals were housed prior to experiments versus during experiments. A graphical overview of the time-line and inter-test intervals should be provided in the main text rather than just as a Supplementary figure.
- Behavioural assessments: if the second trial of the dominance test only started at 13:30, experiments cannot have been conducted between 6:00 and 13:30. Please specify.
- Lines 152-153: it would be important to know when lights were switched on since zebra finches generally do not feed during the night. Did birds that were food-deprived in the morning have a chance to feed before? Otherwise the motivation to feed will be different in birds food-deprived in the morning versus at noon.
- Line 185: video S6 (obstinacy) is missing from the ESM files
- Line 190: We carried out the first round of tests in 2016.
- Lines 191-193: here it would be helpful to have an estimate of the average life span and how much these 2 years actually capture.
- Lines 194-196: repetition from before
- Videos S7, S8, S9 are also missing it seems
- Statistical analyses:
 - o "repeatability" is a population-level parameter and cannot be measured within individuals
 - o The description of repeatability needs some clarification. Were different models calculated for short-term versus long-term repeatability or were all time-points entered in the same model? The model structure (fixed and random effects) need to be spelled out. For example, was "trial" included as fixed effect or cage/pair ID as random effect?
 - o From the results section it becomes obvious that very different variables (latencies, number of, etc) were tested for repeatability. I highly doubt that a Gaussian distribution would be an adequate fit for all of them. Was model fit checked? Or were alternative families such as poisson assumed for some variables?
 - o Also the part describing associations between personality and problem-solving is confusing. First, the authors say that correlations were determined but then they describe models that rely on regression (i.e. directed) rather than correlational (i.e. undirected) associations.
 - o Mixed effects models that rely on a Gaussian error distribution are not "generalized" mixed models.
 - o For all models, the model structure needs to be spelled out in more detail.
 - o How were model assumptions investigated?

Results:

- Line 273: delete "within"
- Lines 290-298: I wonder why repeatability of problem-solving was not assessed. The meta-analysis (Dougherty & Guilette, 2018) that proposed to use multiple measurements for personality traits also emphasised the need to proof that cognitive measures represent consistent individual traits.
- Line 298: "model selection" is not a very meaningful header, something like personality-cognition associations would be more meaningful

- Lines 299-303: “included those behaviours that were repeatable over long-term”: but how exactly? Was the average behavioural value included or the first measurement or all? In addition: Even though these measures come from different tests, they may be correlated and hence violate model assumptions. This would need to be checked and reported.
- Lines 303-304: this info should be presented in the statistical analysis already
- Including animal ID as a random effect probably means that all repeated measurements per individual were included into the model, correct? If so, then how were missing values (e.g. the dominance test only had short-term repeats but not long-term repeats) dealt with? By default, rows containing missing values would not be considered by default as far as I know.
- I suggest to standardise personality estimates to a mean of zero and a comparable variance to ease interpretation of results
- Lines 304-309: now I’m completely lost. Why were separate models calculated for all three problem-solving tasks? If the goal was to estimate the effect of personality traits on problem-solving as a general cognitive trait, then they should be combined into one model. In addition, if only one problem-solving estimate per individual was included a response variable at a time, what does a random effect of ID tell us?
- Line 305 and table 2 SI: spell out the effect please. In addition, from table 2 it is not clear whether the presented effect size relates to the model or to the model comparison. To interpret the biological meaning of the association, it would be necessary to have an effect size for the model (and the specific variable if more than one would be included in the model). In addition, to determine the biological significance of findings, an estimate of model fit would be helpful. See for example “Nakagawa, S., & Schielzeth, H. (2013). A general and simple method for obtaining R² from generalized linear mixed-effects models. *Methods in ecology and evolution*, 4(2), 133-142.”

Discussion

- Given my doubts on several of the statistical analyses, I will not review the discussion in detail at this stage. Just three general comments:
- If the main objective was to test for personality-cognitions associations as suggested by the background and the title, then I would expect the main focus of the discussion to be on that too. Currently, the main focus is in discussing the single behavioural/ personality traits and why or why not a long-term repeatability was found.
- The whole discussion is quite speculative which makes it hard to read and, even harder to pinpoint what exactly we have learned from this study. Where does it close some gaps in the literature?
- The discussion is very focused on zebra finches. I am missing the broader biological context in terms of a) how do findings relate to findings in other species and b) the general scientific hypothesis underlying the study
- The conclusion paragraph is too weak, it only reiterates what has been found in the study once again.

Appendix B

Dear Dr. Mathot and Dr. Padian,

We would like to thank you and the reviewers for your positive feedback on our study design and dataset, and for your detailed and constructive feedback on our manuscript (RSOS-210413). We are grateful for the opportunity to resubmit our substantially revised manuscript. Please find below a point-by-point response (in bold) to the comments provided by you and the reviewers, with line number references to our *Track Changes PDF* document. In nearly every case, we have implemented the suggested changes, and we feel the paper has significantly improved as a result. We hope that you agree.

Thank you for considering our revised manuscript for publication.

Sincerely,

Lisa P. Barrett

Corresponding Author, on behalf of all co-authors

Dear Dr Barrett

The Editors assigned to your paper RSOS-210413 "Links between personality traits and problem-solving performance in zebra finches (*Taeniopygia guttata*)" have made a decision based on their reading of the paper and any comments received from reviewers.

Regrettably, in view of the reports received, the manuscript has been rejected in its current form. However, a new manuscript may be submitted which takes into consideration these comments.

We invite you to respond to the comments supplied below and prepare a resubmission of your manuscript. Below the referees' and Editors' comments (where applicable) we provide additional requirements. We provide guidance below to help you prepare your revision.

Please note that resubmitting your manuscript does not guarantee eventual acceptance, and we do not generally allow multiple rounds of revision and resubmission, so we urge you to make every effort to fully address all of the comments at this stage. If deemed necessary by the Editors, your manuscript will be sent back to one or more of the original reviewers for assessment. If the original reviewers are not available, we may invite new reviewers.

Please resubmit your revised manuscript and required files (see below) no later than 25-Nov-2021. Note: the ScholarOne system will 'lock' if resubmission is attempted on or after this deadline. If you do not think you will be able to meet this deadline, please contact the editorial office immediately.

Please note article processing charges apply to papers accepted for publication in Royal Society Open Science (<https://royalsocietypublishing.org/rsos/charges>). Charges will also apply to papers transferred to the journal from other Royal Society Publishing journals, as well as papers submitted as part of our collaboration with the Royal Society of Chemistry (<https://royalsocietypublishing.org/rsos/chemistry>). Fee waivers are available but must be requested when you submit your manuscript (<https://royalsocietypublishing.org/rsos/waivers>).

Thank you for submitting your manuscript to Royal Society Open Science and we look forward to receiving your resubmission. If you have any questions at all, please do not hesitate to get in touch.

on behalf of Dr Kimberley Mathot (Associate Editor) and Kevin Padian (Subject Editor)
openscience@royalsociety.org

Thank you! We are so glad you think the topic will be of interest to your readership and that after appropriate revision, the manuscript may be publishable in *Royal Society Open Science*.

Associate Editor Comments to Author (Dr Kimberley Mathot):
Comments to the Author:
Dear Dr. Barrett,

Thank you for submitting your article for consideration at Royal Society Open Science. Your paper has now been reviewed by three experts in the field. Each of them agree that the study design and data set are solid. However, each of the referees also raised significant concerns about the appropriateness of the analyses that were conducted, specifically in reference to the stated objectives. Because I believe addressing the referee concerns will involve substantial work and may dramatically alter the conclusions drawn from the study, I am recommending the manuscript be rejected with the possibility to resubmit.

I felt that the referee reports were particularly detailed and constructive, and I hope you will consider their comments and revise accordingly. I believe that the manuscript will be much stronger if you are able to address their points. In particular, the misalignment between the concept of cognitive syndromes (the outcome of co-evolution of behavioural and cognitive traits) and your current analytical approach (causal effect of behavioural traits on problem-solving) should be addressed. Structural equation modelling may be a more appropriate framework, or multivariate analyses that estimate trait covariances without assuming causality.

Thank you for the opportunity to revise and resubmit our manuscript and for the extension of the resubmission deadline. We greatly appreciate your helpful insights and believe the manuscript is much stronger now that we have incorporated your and the reviewers' suggestions.

Thank you for your feedback and for suggesting new analysis techniques. We have looked into your suggestions but see the following problems:

- Causality is not an assumption of regression (e.g., Wheaton & Young, 2021), and we do not assume causality here. Regression can only indicate a correlation per se (e.g., Gelman & Hill, 2007), and is not used to establish causality on its own. Since we do not assume causality, we do not feel multivariate analyses that estimate covariance without assuming causality would be more appropriate than the regression we used.
- The regression methodology we employ has been commonly used in the animal personality and cognition literature (e.g., Cole & Quinn, 2012; Guillette et al., 2011; Martins et al., 2007; O'Shea et al., 2017; Zandberg et al., 2017), as well as in the human personality literature (e.g., Paunonen & Ashton, 2001).
- We considered using structural equation modeling, but unfortunately there is not enough background literature to make informed predictions concerning how the specific responses measured here interact with one another. Also, it is our understanding that structural equation modeling *is* itself a type of correlation/regression (e.g., Schumaker & Lomax, 2004), and so we believe our method of using regression is a valid one.

Therefore, we do not think those techniques are appropriate for our dataset and questions. Additionally, we agree that some of our language suggested that we were inferring causal relationships in the original manuscript, but the regression technique that we used does not actually assume causality and is commonly used in the personality and cognition literature when investigating similar questions to those in our study. We have therefore changed the language in our revised manuscript to clarify that we are not assuming causal relationships between variables (e.g., Lines 50, 350, 389, 402, 487, etc.), but we have retained our regression analysis.

Cole, E.F. & Quinn, J.L. (2012). Personality and problem-solving performance explain competitive ability in the wild. *Proc. R. Soc. B*, 279, 1168-1175.

Gelman, A. & Hill, J. (2007). *Data analysis using regression and multilevel/hierarchical models*. Cambridge: Cambridge University Press.

Guillette, L.M., Reddon, A.R., Hoeschele, M., & Sturdy, C.B. (2011). Sometimes slower is better: slow-exploring birds are more sensitive to changes in a vocal discrimination task. *Proc. R. Soc. B*, 278, 767-773.

Martins, T.L.F., Roberts, M.L., Giblin, I., Huxham, R., & Evans, M.R. (2007). Speed of exploration and risk-taking behavior are linked to corticosterone titres in zebra finches. *Hormones and Behavior*, 52, 445-453.

O'Shea, W., Serrano-Davies, E., & Quinn, J.L. (2017). Do personality and innovativeness influence competitive ability? An experimental test in the great tit. *Behavioral Ecology*, 28(6), 1435-1444.

Paunonen, S.V. & Ashton, M.C. (2001). Big five factors and facets and the prediction of behavior. *Journal of Personality and Social Psychology*, 81, 3, 524-539.

Schumacker, R.E. & Lomax, R.G. (2004). *A beginner's guide to structural equation modeling*. 2nd Edition, Lawrence Erlbaum Associates, Mahwah.

Young, M. & Wheaton, B. (2020). *Generalizing the Regression Model: Techniques for Longitudinal and Contextual Analysis*. United States: SAGE Publications.

Zandberg, L., Quinn, J.L., Naguib, M., & van Oers, K. (2017). Personality-dependent differences in problem-solving performance in a social context reflect foraging strategies. *Behavioural Processes*, 134, 95-102.

In addition to the referee comments, I have a few further points I would like to see addressed in a revision.

1) estimation of short versus long-term repeatability: there is no need to take averages from each time category. In fact, this can lead to substantial bias in repeatability estimates. Instead, you should consider using the method outlined by Araya-Ajoy et al. 2015. As you are not estimating individual reaction norms, it will require slight modification of the model parameterization from what is provided in that paper (e.g., including series, but not series|ID) as random effects.

Araya-Ajoy, Y. G., Mathot, K. J. and Dingemanse, N. J. (2015). An approach to estimate short-term, long-term and reaction norm repeatability. *Methods in Ecology and Evolution* 6, 1462-1473.

Thank you for this suggestion. We have revised our analyses for Latency to Feed (Novel Object), which was repeated in both years (and hence was averaged for long-term repeatability in our original analysis). We also revised our analysis for number of movements (General Activity Test) (which had two trials in both years) following Araya-Ajoy et al. (2015) (Lines 359-362). The other tests were conducted only once each year and so only long-term repeatability could be calculated for those tests. We found that General Activity was no longer significantly repeatable over the long term after this revision. As a result, one of our top models linking personality to problem-solving success changed (Average Latency to Solve) (e.g., Table 5).

2) Acknowledgement that your long-term repeatability involves both a long time interval (2 years), but also a change in context. My understanding is that birds were housed with mates only during the second set of measures (that is why you couldn't assess dominance over the long-term). This could result in lower long-term repeatability due to IxE (individual differences in reaction norms) in response to the changes in housing conditions rather than simply time. There isn't anything you can do to address this confound, but it should at least be acknowledged.

This is a great point. We have added a note about this to our discussion (Lines 512-513).

3) Please provide the rationale for key analytical decisions in the main text, not the ESM.

We have moved SI Table 1 (which provides rationale for exclusion of certain traits) to the main text (Table 2). We hope this clarifies our key analytical decisions, but please let us know if we should add additional information.

Reviewer comments to Author:

Reviewer: 1

Comments to the Author(s)

The present study assesses links between different personality traits and three different problem-solving tasks in captive zebra finches. Several personality traits predicted success in

the problem-solving tasks but some associations were task-specific.

The manuscript is overall well written and addresses a relevant question that has recently received increasing attention; that is, whether personality traits and cognitive abilities are correlated. While the question is not new, the authors are among the firsts to assess such links across different problem-solving tasks.

Thank you for taking the time to review our manuscript. We are glad you think it is well-written and timely!

I have several rather minor comments that I would like to see addressed.

L61-71: I would wish for an explanation as to why it would be important to understand the link between personality and cognition.

We have added an explanation about the importance of this link (Lines 74-76: “Understanding the links between personality and cognition contributes to our understanding of the coevolution of behavioural responses; determining patterns between seemingly disparate traits can help predict how animals will respond to novel stimuli or changes to their environment.”).

L73: Please change to ‘problem’.

We changed this to “problem” (Line 80, 117).

L83: It says ‘typically’ – which studies are the exception? Please mention those here.

We have removed “typically” (Line 129).

L85: Please clarify which definition of personality is used. According to the abstract, personality differences are ‘consistent differences across time or contexts...’, suggesting they can be single traits; here it is suggested that they are ‘multidimensional collection of behavioural traits’. Would be ‘syndromes’ more appropriated here?

Yes, thank you. We changed this to “behavioral syndrome” (Lines 98-99, 131).

L91 onwards: Several studies have already assessed personality traits across longer periods in zebra finches; these should be acknowledged here (e.g. 7 months: David et al. 2012; 3 months: Schuett et al. 2011; 1 year: Wuerz et al. 2015).

Yes, we agree. We have added these citations here (Lines 109-110).

L131: I suggest using degree Celsius instead of Fahrenheit.

We converted to Celsius (Lines 192-193).

L 155: Could you give a more complete list of physical interactions that were used, please?

We have listed the physical interactions used (Lines 220-221).

L193: Were birds tested in the same groups across the years?

The only group test was the dominance test (now relabeled as Assessment of Dominance Hierarchies), which was not repeated in 2018/across the years (please see lines 274-275, 278-280). Birds were tested in the same groups across trials; we added this detail (Lines 225-226).

L223: It seems I do not have access to video 7-9.

We apologize for this. We are not sure why these videos did not play (anyone with the link should have access). Please let us know if you have any difficulty playing them now.

- SI Video 1: <https://youtu.be/6Lzifoa9Eyk>
- SI Video 2: <https://youtu.be/Pmv7JserD-Y>
- SI Video 3: <https://youtu.be/OkfPM30rwPA>
- SI Video 4: <https://youtu.be/n6YHfUq3yEQ>
- SI Video 5: <https://youtu.be/n9H7qHCoVQg>
- SI Video 6: <https://youtu.be/X3LEYBUubQs>
- SI Video 7: <https://youtu.be/uTIPmxunu1s>
- SI Video 8: <https://youtu.be/0pLh6WNt4b4>
- SI Video 9: <https://youtu.be/TKAVXMAMTWQ>

L259: Are the patterns / repeatability values similar if only the data of the first trial 2016 and 2018 are used? As repeatabilities in the short term are not that high, it seems more appropriate not to use the means.

We did look into this possibility, and found that including the first trial of 2016 and 2018 did not change our results. However, our analyses have now changed following a suggestion from the editor, and so we no longer use averages of 2016 and averages of 2018 in our long-term repeatability analyses (Lines 360-363).

Analyses/Results: Potential sex-differences in patterns have not been assessed, even though some studies have shown that repeatabilities of personality traits significantly differ between males and females and that links between personality and cognitive abilities can be sex-specific (for latter, see also intro of current study). Even though the sample size is not huge, I suggest considering such potential sex differences in the analyses.

Sex differences were not a focus of the current study. However, we have tested sex differences in repeatability of personality and did not find any significant sex differences. We have added this to the manuscript (Line 363-364, 412-420).

L265: Were model assumptions checked and met? Usually, latencies do not follow Gaussian error structures. Please clarify in the text.

Yes, thank you. We have checked model assumption (distribution of residuals), log (x+1) transformed non-normal data, and applied a Poisson or binomial distribution where appropriate. We have clarified this in the text (Lines 358-360, 377-382).

Line 298-304: It is unclear if those are the maximal models or reduced models; if the first, it seems more appropriate to add this info to the stats section.

These are the maximal models that were based on which traits were found to be significantly repeatable (in the Behavioural Assessments subsection). Since the models were dependent on our repeatability results, we felt it would not be appropriate to list which traits were repeatable in the statistical analysis section (i.e., that they belong in the results section). However, we have moved this to the Statistical Analysis section (Lines 382-387, 431-436). Please let us know if anything is unclear.

L304-313: I suggest adding the direction of effect, especially for those results that are not summarised in Fig. 3 captions.

We have added the direction of the effects (Lines 436-449).

Discussion: the section on repeatability of behaviour is quite long and could be shortened considerably.

We are careful to be thorough in our discussion of our repeatability results, as this was a primary objective of our study. But, we recognize your concern, and we are happy to delete Lines 482-507 if the editor feels it would be helpful.

SI Document 1: I suggest adding this information to Fig 2 and deleting this document here.
We have added the dimensions to Fig. 2 and deleted the SI Document.

References:

David, M., Auclair, Y., & Cezilly, F. (2012). Assessing Short- and Long-Term Repeatability and Stability of Personality in Captive Zebra Finches Using Longitudinal Data. *Ethology*, 118(10), 932-942. doi: 10.1111/j.1439-0310.2012.02085.x

Schuett, W., Dall, S. R. X., & Royle, N. J. (2011). Pairs of zebra finches with similar 'personalities' make better parents. *Animal Behaviour*, 81(3), 609-618. doi: 10.1016/j.anbehav.2010.12.006

Wuerz, Y., & Krüger, O. (2015). Personality over ontogeny in zebra finches: long-term repeatable traits but unstable behavioural syndromes. *Frontiers in Zoology*, 12(1), 1-14.

Reviewer: 2

Comments to the Author(s)

Please see all comments in the attached file (inserted here)

Manuscript RSOS-210413 investigates associations between several personality traits such as dominance, activity or neophobia and problem-solving ability in zebra finches. In addition, the authors emphasize their estimates of long-term repeatability estimates. In general, the topic of the study is timely and will likely attract many readers. The number of animals tested and the design of experiment is sound and carefully chosen based on existing literature. Thus, I am convinced that the study offers a valuable dataset which will make a good contribution to the literature. In its current state however, I cannot recommend the manuscript for publication. Mainly, I have serious concerns about the statistical analysis carried out and therefore in the interpretation of results. In addition, methods are often described superficially which makes it hard to judge their appropriateness. Given the potential merit I see in the data, I would be happy to review a substantially revised version of the article.

Thank you for your constructive feedback on our manuscript. We are glad you see merit in our data and feel the study is timely. We appreciate your willingness to review our revised article.

General comments:

The authors place the study into the area of cognitive syndromes in the background but later test for the predictive effects of personality traits on cognitive traits. There are conceptual differences between these two approaches. The concept of a cognitive syndrome relies on correlations between personality and cognitive traits (for example because both might have been shaped by correlational selection or through pleiotropic effects). Thus, correlations (i.e. without a priori assumptions of predictive effects) should be used (and here, following the concept of behavioural syndromes, assumptions primarily apply to between-individual rather than phenotypic correlations). The statistical methods applied here on the other hand, assume that variation in personality traits predict, i.e. generate variation in cognitive traits. I suggest to apply a multivariate mixed-effect modelling approach throughout. The statistical analyses are

hard to follow and seem inappropriate in several places. Therefore, conclusions and discussion cannot be reliably reviewed at this stage.

We agree with you about the conceptual differences between cognitive syndromes and predictive effects of personality traits on cognitive traits. Causality is not an assumption of regression (e.g., Wheaton & Young, 2021), however, and we do not assume causality here. Please see our response to the editor about this above. Please also see our specific revisions below.

Background:

- I am missing a part that explains why it is important to study the relationship between personality and cognition in a biological context.

In addition, a short introduction on why these seemingly different traits should be associated is missing. "Individual variation in cognition [...] likely relates to personality types" is a bit short.

We have added detail about the importance of this study ("Understanding the links between personality and cognition contributes to our understanding of the coevolution of behavioural responses; determining patterns between seemingly disparate traits can help predict how animals will respond to novel stimuli or changes to their environment" (Lines 74-76)).

- To understand why specific personality measures were chosen to be tested here, it would be necessary to know that the scientific hypothesis behind the expected association between certain cognitive traits and certain personality, relies on a connection of both types of traits to a shared risk-reward trade-off. I can for example not understand why general activity should be measured since I fail to see the connection to a risk-reward trade-off. To prevent more studies reporting mixed results in the future, the clear connection to the scientific background should always be made crystal clear. Since we have well-developed scientific hypotheses that can be tested, we should be past the stage where more or less random trait associations can be reported.

Our predicted associations between personality traits and innovative performance are not random but have not been tested before in zebra finches. We therefore based our predicted associations on the available relevant literature, as cited in Table 1. For example, activity has been measured with regard to predator-prey tradeoffs, where active individuals might be prone to inappropriate activity in the presence of predators; i.e., higher activity results in higher feeding rates, but also higher predation risk (Sih et al., 2004). In terms of cognition, more proactive individuals are faster at initial learning but slower to relearn when contingencies have changed. In other words, faster individuals more readily form routines but react slowly to changes in their environment, so making fast decisions might be beneficial over the short-term as individuals gain more resources, but these quick decisions may be inaccurate or risky (Guenther et al., 2014). We have added this explanation to the text (Lines 171-177).

- The connection between innovative problem-solving and cognition as a whole is not introduced.

We have added a connection between innovative problem solving and cognition. Please see lines 78-86, including an acknowledgement about the debate regarding whether innovation is an indicator of cognitive ability (following a suggestion from Reviewer 3).

Methods:

- Lines 121-128: The timeline is a bit unclear. The authors say testing started 4 months after obtaining birds from the breeder but a few lines further down, a two-week habituation period is mentioned. How do these relate to each other? In addition, it is not fully clear in which cages/

group sizes animals were housed prior to experiments versus during experiments. A graphical overview of the time-line and inter-test intervals should be provided in the main text rather than just as a Supplementary figure.

We have now added detail about this: the habituation period is in relation to the flight cage group housing (Line 187). Please see Lines 184-186 for detail about the cages animals were housed in prior to/during experiments. We have moved the supplemental figures into the main text – please see Figures 3 and 4.

- Behavioural assessments: if the second trial of the dominance test only started at 13:30, experiments cannot have been conducted between 6:00 and 13:30. Please specify.

Thank you for pointing out this error in the main text. We have changed 13:30 to 13:45 (Line 198).

- Lines 152-153: it would be important to know when lights were switched on since zebra finches generally do not feed during the night. Did birds that were food-deprived in the morning have a chance to feed before? Otherwise the motivation to feed will be different in birds food-deprived in the morning versus at noon.

Yes, this is a great point. Birds food-deprived in the morning did have a chance to feed before, as the lights came on an hour before deprivation began. We have added this information (Lines 218-219).

- Line 185: video S6 (obstinacy) is missing from the ESM files

Apologies. Please let us know if you have any trouble accessing it now:

SI Video 1: <https://youtu.be/6Lzifoa9Eyk>

SI Video 2: <https://youtu.be/Pmv7JserD-Y>

SI Video 3: <https://youtu.be/OkfPM30rwPA>

SI Video 4: <https://youtu.be/n6YHfUq3yEQ>

SI Video 5: <https://youtu.be/n9H7qHCoVQg>

SI Video 6: <https://youtu.be/X3LEYBUubQs>

SI Video 7: <https://youtu.be/uTIPmxunu1s>

SI Video 8: <https://youtu.be/0pLh6WNt4b4>

SI Video 9: <https://youtu.be/TKAVXMAMTWQ>

- Line 190: We carried out the first round of tests in 2016.

We have revised this sentence (Line 274).

- Lines 191-193: here it would be helpful to have an estimate of the average life span and how much these 2 years actually capture.

We have added this information; zebra finches have been estimated to live 1.3-5 years in the wild, and 5-9 years in captivity (Lines 277-278).

- Lines 194-196: repetition from before

Here (Lines 280-284) we are specifying why certain tests were not repeated multiple times (unlike the other tests, which were repeated across trials) *within* 2016 and *within*

2018. Earlier (Lines 274-277), we explain why we chose to test across two years as our long-term test-retest interval. These lines are thus different in meaning from what was stated above. We have attempted to further clarify this in the revision. Please let us know if this does not address your comment about repetition from “before.”

- Videos S7, S8, S9 are also missing it seems

We are not sure why these videos did not play (anyone with the link should have access). Please let us know if you have any difficulty playing them now.

SI Video 1: <https://youtu.be/6Lzifo9EyK>

SI Video 2: <https://youtu.be/Pmv7JserD-Y>

SI Video 3: <https://youtu.be/OkfPM30rwPA>

SI Video 4: <https://youtu.be/n6YHfUq3yEQ>

SI Video 5: <https://youtu.be/n9H7gHCoVQg>

SI Video 6: <https://youtu.be/X3LEYBUubQs>

SI Video 7: <https://youtu.be/uTIPmxunu1s>

SI Video 8: <https://youtu.be/0pLh6WNt4b4>

SI Video 9: <https://youtu.be/TKAVXMAMTWQ>

- Statistical analyses:

o “repeatability” is a population-level parameter and cannot be measured within individuals

We completely agree with you and apologize for the error. We have deleted this phrase (Line 354).

o The description of repeatability needs some clarification. Were different models calculated for short-term versus long-term repeatability or were all time-points entered in the same model? The model structure (fixed and random effects) need to be spelled out. For example, was “trial” included as fixed effect or cage/pair ID as random effect?

We have clarified the description of our repeatability analyses (Lines 335-340).

o From the results section it becomes obvious that very different variables (latencies, number of, etc) were tested for repeatability. I highly doubt that a Gaussian distribution would be an adequate fit for all of them. Was model fit checked? Or were alternative families such as poisson assumed for some variables?

We have added more clarification on repeatability response distributions to our statistical analysis section. We checked model fits by examining the distribution of residuals, and we $\log(x+1)$ transformed non-normal latency responses to ensure a Gaussian distribution was appropriate. Poisson was assumed for count data. Our rate response (Mirror Interactions) was rounded to a whole number, and a Poisson distribution was used for these count data (Lines 358-363).

o Also the part describing associations between personality and problem-solving is confusing. First, the authors say that correlations were determined but then they describe models that rely on regression (i.e. directed) rather than correlational (i.e. undirected) associations.

We used both Pearson correlations and regression (which is also a correlational method) in our analyses. We have revised the description of our statistical methods to clarify this (Lines 375-377) and to avoid using directed language (Lines 378, 380-381, and throughout). Please also see our response to the editor about our regression analyses.

o Mixed effects models that rely on a Gaussian error distribution are not “generalized” mixed models.

Thank you for pointing this out error. We removed “generalized” (Line 380).

o For all models, the model structure needs to be spelled out in more detail.

These are the maximal models that were based on which traits were found to be significantly repeatable (in the Behavioural Assessments subsection). Since the models were dependent on our repeatability results, we felt it would not be appropriate to list which traits were repeatable in the statistical analysis section (i.e., that they belong in the results section). However, we have moved this to the Statistical Analysis section (Lines 382-387, Lines 431-436). Please let us know if anything is unclear.

o How were model assumptions investigated?

We investigated the distribution of the residuals to determine appropriate response distributions. We assumed Poisson distributions for count data. We have added this information to our Statistical Analysis section (Lines 358-360).

Results:

- Line 273: delete “within”

We have deleted “within individuals” as suggested (Line 393).

- Lines 290-298: I wonder why repeatability of problem-solving was not assessed. The metaanalysis (Dougherty & Guillette, 2018) that proposed to use multiple measurements for personality traits also emphasised the need to proof that cognitive measures represent consistent individual traits.

Repeatability of problem-solving performance was not an original focus of the study though we agree that it is worth investigating. We have added a test of repeatability of latency to solve. We found a trend toward latency to solve being consistent across tasks ($r = 0.387$, $n = 41$, $p = 0.052$). We have added this to our results (Lines 428-429).

- Line 298: “model selection” is not a very meaningful header, something like personality cognition associations would be more meaningful

Thank you for this suggestion. We have changed our header to “Personality-cognition associations” (Line 430).

- Lines 299-303: “included those behaviours that were repeatable over long-term”: but how exactly? Was the average behavioural value included or the first measurement or all? In addition: Even though these measures come from different tests, they may be correlated and hence violate model assumptions. This would need to be checked and reported.

The average behavioral value was included (Line 375, 384-387). Also, we checked for correlations and report which measures were correlated in Table 2. None of these retained measures were correlated across tests (Lines 367-368).

- Lines 303-304: this info should be presented in the statistical analysis already

We removed this information because we did not include ID as a random effect here, as there was only one trial per animal for each task (as you helpfully point out below) (Line 387; Lines 435-436).

- Including animal ID as a random effect probably means that all repeated measurements per individual were included into the model, correct? If so, then how were missing values (e.g. the dominance test only had short-term repeats but not long-term repeats) dealt with? By default, rows containing missing values would not be considered by default as far as I know.

Thank you for this query. We did not include ID as a random effect, as there was only one trial per animal for each task. We used the short-term average for the dominance test (now relabeled as Assessment of Dominance Hierarchies as per Reviewer 3's suggestion) alongside behaviors that were repeatable over 2 years (Please see lines 384-387).

- I suggest to standardise personality estimates to a mean of zero and a comparable variance to ease interpretation of results

It is our understanding that standardizing estimates is needed when comparing effects of independent variables on the response (e.g., Cumming, 2014). Since we are not comparing effect sizes, we would prefer to use unstandardized estimates, but we are happy to standardize them if the editor deems it necessary.

Cumming, G. 2014. The New Statistics: Why and How. *Psychological Science*, 25,1, 7–29.

- Lines 304-309: now I'm completely lost. Why were separate models calculated for all three problem-solving tasks? If the goal was to estimate the effect of personality traits on problemsolving as a general cognitive trait, then they should be combined into one model. In addition, if only one problem-solving estimate per individual was included a response variable at a time, what does a random effect of ID tell us?

We do not assume that all of the tasks measured a general cognitive trait, so we treated all tasks separately in case they were measuring slightly different aspects of cognition. We have added a note about this to our statistical analysis section (Lines 379-380). Also, we removed our erroneous note about having a random effect of ID, as each animal received only one trial per task (Line 387; Lines 435-436). We have added a test of repeatability of latency to solve. We found a trend that latency to solve was consistent within individuals across tasks ($r = 0.387$, $n = 41$, $p = 0.052$). We have added this to our results (Lines 428-429).

- Line 305 and table 2 SI: spell out the effect please. In addition, from table 2 it is not clear whether the presented effect size relates to the model or to the model comparison. To interpret the biological meaning of the association, it would be necessary to have an effect size for the model (and the specific variable if more than one would be included in the model). In addition, to determine the biological significance of findings, an estimate of model fit would be helpful. See for example "Nakagawa, S., & Schielzeth, H. (2013). A general and simple method for obtaining R^2 from generalized linear mixed-effects models. *Methods in ecology and evolution*, 4(2), 133-142."

Please see Table 1 SI for effect sizes for specific variables and confidence intervals, and please see Table 5 for R^2 values of the top models. We have added information about the direction of effects to our results section (Lines 435-449).

Discussion

- Given my doubts on several of the statistical analyses, I will not review the discussion in detail at this stage. Just three general comments:

- If the main objective was to test for personality-cognitions associations as suggested by the background and the title, then I would expect the main focus of the discussion to be on that too. Currently, the main focus is in discussing the single behavioural/ personality traits and why or why not a long-term repeatability was found.

Our main objectives were two-fold (Please see lines 165-167). We have added more about the significance of our results to the Discussion (see below).

- The whole discussion is quite speculative which makes it hard to read and, even harder to pinpoint what exactly we have learned from this study. Where does it close some gaps in the literature?

- The discussion is very focused on zebra finches. I am missing the broader biological context in terms of a) how do findings relate to findings in other species and b) the general scientific hypothesis underlying the study

- The conclusion paragraph is too weak, it only reiterates what has been found in the study once again.

Thank you for this constructive critique. We have added more discussion about the biological importance of our study and where our study closes some gaps in the literature (e.g., Lines 479-491, 495, 500-501, 521-526, 532-533, 547-550, 563-566, 575-576).

Reviewer: 3

Comments to the Author(s)

This study examines 1) the long-term repeatability of personality traits, and 2) whether multiple cognitive tests correlate with personality measures. A battery of personality tests (dominance test, general activity measurement, novel object test, startle test, struggle test, mirror-interaction test) and a battery of problem-solving tests (string-pull test, wire-removal test, and a lid-flip test) were conducted on 41 captive zebra finches. Personality tests were repeated after a short interval, and partly repeated after two years. Results showed that most variables measured in the personality tests were consistent over a two years period. The success in problem-solving tests was best predicted by dominance and boldness, with less dominant but bolder birds more likely to solve a higher number of tests.

I think this is a very nice and thorough study, with the potential to advance scientific knowledge in the field. However the presentation of this work will need extensive revisions. Unfortunately, I think that the manuscript, data analyses and result interpretation all need further work before being ready for publication. The overall aim of the study does not become apparent yet, the take away message remains unclear and diffuse, the story still needs shaping, and conclusions seem at least disconnected from the results. In short, I do not think the authors do justice to the impressive amount of effort that went into this data collection.

We appreciate your time and helpful comments on our manuscript. We are glad you found that the study was thorough and could advance the field. We believe the manuscript is much stronger after incorporating your edits.

General comments

From the way it is currently presented, it is unclear whether the authors consider this a methodological study, highlighting potential and difficulties of measuring multiple personality traits in relation to cognition, or actually a study investigating this connection and its meaning for

cognitive and behavioural evolution and research. The two levels are constantly mixed. If this is a study whose results are intended to contribute to the understanding of individual variation in cognition (and I think it definitely has the potential to do this), there is no clear reasoning on why it was attempted, and what the obtained results mean in a biological sense. The results illustrate that there is a connection between some personality measures and problem solving, but why is the presence of this connection considered proof that running multiple tests is important? I do agree that this is what results show, and also that this methodological approach is important, but I do not see an evident connection in how this is presented here. It would be good if the authors clarified what they think is the main point of this paper, a validation of a methodological approach or the biological relevance of this link. This will help to focus more clearly on the aspects they want to highlight, and to create a hierarchical order of importance of their findings in the narrative that will be clear to readers as well.

Yes, the study was an investigation of the connection between personality and cognition, where personality was measured over two years. We consider the methodological approach as being integral to the biological relevance, and these two levels are therefore mixed. We see it as an advantage that our paper has multiple yet complementary goals. We have clarified about the importance of the connection between personality and problem solving and the importance of running multiple tests (e.g., “Collecting multiple measures of personality would help to identify how individual differences in personality relate to individual differences in response to varied environmental contingencies.” (Lines 96-98)). We have also reorganized our introduction to more clearly explain the different objectives of the paper (Lines 78-177).

My second issue concerns the claim that multiple cognitive traits were assessed in combination with multiple measures of personality. The problem-solving tests were repeated (great!), and even though it is not clear if a similar assessment was carried out in 2018, this is a very important aspect that has all my approval. However, while the personality tests indeed address multiple aspects of animal personality, the three cognitive tasks are variations of the same test, a removal of something to access the food reward visible through a transparent barrier. The three tasks were targeting the same cognitive ability, and relied on similar cognitive mechanisms (L234). The variation was in the non-cognitive aspects such as motor actions required to solve them.

We completely agree with you. Here we do not make claims about having measured multiple cognitive traits, however. Instead, we state that we measured problem-solving performance on three different tasks, which each required different motor patterns to solve (please see, for example, Line 165). After all, it is interesting to test one problem-solving ability in multiple ways in order to look at validity of the tasks, and here we wanted to focus on one cognitive ability for this reason. We also added some clarification about this (see below). We also added an analysis to test for repeatability of problem-solving ability across the three tasks (Lines 428-429).

The papers that the authors cite (e.g. Griffin et al., 2015) in support of this approach, actually recommend testing multiple cognitive traits along with their repeatability and other personality traits. While the authors often write “three problem-solving tests”, mentions of repeated tests in the context of the need for multiple cognitive assessments are misleading. My recommendation is that the authors clarify that the multiple traits measured here only refer to personality (to avoid overselling) and, more importantly, that they explain why they think is important multiple measures of personality in connection with a single, albeit repeated, cognitive task. At the moment the reader is left with the impression that the main reason is that some measures of personality correlate with the problem-solving performance and others are not, without going any deeper into the implications of this connection.

Thank you for this suggestion. We treated each task separately in case they were measuring slightly different aspects of cognition, even though we did not intend them to. However, we have added clarification that the multiple traits measured here only refer to personality to avoid overselling (Lines 41, 160, 165). We have also added an explanation on why we think multiple measures of personality are important to investigate in connection with innovative tendency (Lines 51-53, 96-105, 148-149).

Another aspect that in my opinion requires revision is that, currently, the rationale for this study is barely explained. The authors cite the meta-analysis that found that while personality and cognition are related, the direction of this relationship is not the same across species, or tasks etc. From the way this is presented, I wonder if we should expect any kind of biological relationship to be so universal and unidirectional, and why the species- or task-dependence is considered problematic. Personally, I think it is great that further studies are conducted on the topic – I just could not understand what are the authors ideas on this, though, as they are not presented in the text beyond the simple statements that the relationship not always goes in the same direction and that, for this reason (?), multiple (personality) traits should be considered. This intellectual contribution in contextualising findings is, in my opinion, quite as necessary as robust results themselves.

We have clarified our position that multiple measures are needed because the relationship is often muddled based on how a trait is measured (i.e., what assessment is used, or which measures are extracted from the assessment)—not because there is no universal relationship (Lines 51-53, 89-90, 96-105, 148-149).

Finally, I think that the thoroughness with which the personality tests were conducted should extend to the analysis of the data derived from them. The various tests yielded a wealth of behavioural observations on several variables – but of course not every single variable measured in, for example, a neophobia test is an indicator of neophobia, and most likely expresses a combination of various traits. Currently, all variables are presented as carrying the same significance and being indicators of only the personality trait after which the corresponding test is named. However, the study of animal personality requires a more critical approach towards the measured variables, and a careful interpretation and evaluation of the observed responses. Have these tests been validated in an ecological setting before concluding that they actually measure boldness, aggressiveness, etc.? If not, what does it tell us that an individual approaching a mirror has a better chance of solving a foraging task? One test can simultaneously be influenced by and thus measure two or more personality traits. Although a test that directly measures a targeted trait may be a desirable goal, in reality a test will likely be influenced by multiple traits at the same time (Carter et al., 2013; Réale et al., 2007). A critical re-evaluation of the variable presented as personality traits is required before this work could be considered ready for publication. On this, I recommend the review by Carter et al. (2013, *Biol. Rev.* (2013), 88, pp. 465–475. 465, doi: 10.1111/brv.12007) as a very valuable starting point to start the revision process.

Although we chose to administer tests that have been widely used in the literature (and have all been used with zebra finches specifically), and that could play a role in innovative problem-solving, we agree with you that it is important to carefully evaluate what these tests are measuring and whether they carry equal significance. We added a discussion of this point, including a caveat about how the tests used throughout the literature are likely influenced by multiple traits at the same time, that some tests may not actually be measuring the trait(s) we think they are measuring, and some traits may carry more weight than others (Lines 515-519) (following Carter et al., 2013).

Specific comments

L36-39: no connection between the first two sentences of the abstract. Why is it important to examine personality and cognition in the same study? I think this info should come before the methodological approaches used in the past, to place this study in the context of current knowledge.

We have added a connection in our abstract (“Links between personality and cognitive ability inform our understanding how behavioural traits coevolve.”; Lines 37-38).

L45-49: From this conclusion it seems that the aim of the study is to support the recommendation of conducting multiple assessments of personality, but it does not address why. Why is it important for the study of cognition to have robust proof that personality is linked to cognition? And why is it important to know under which circumstances and how specific personality traits affect performance in cognitive tests?

We have added more explanation on the importance of obtaining multiple measures (Lines 51-55).

L60-116: In its current form, the introduction is a mixture of unconnected statements about personality being important for fitness, having a non-linear relationship with cognition, and methodological approaches. Considering that personality is the consistent among-individual variation in behavioural responses, I would have expected some mentions on why variation in cognitive performance is important, why we should be concerned with it, and why should we consider the other behavioural aspect of the individuals under study to properly assess the behavioural performance we use to infer cognitive processes and abilities. Instead there are a lot of disconnected sentences that do not reach the heart of the problem, in my opinion, but keep mixing the methodological level with the biological significance of the topics addressed. Of course, this does not detract from the merits of the study, but makes it much harder to appreciate them. I recommend an extensive revision of this section.

We have revised our introduction accordingly (Lines 78-144).

L61-71: the introductory paragraph does not seem to lead clearly to the research question. Why is this link considered relevant? And what were the basis to expect that the relationship between and cognition should be a universal constant, where most behavioural and cognitive aspects are indeed sex-, condition-, age-, motivation-, and context-dependent (L68-70)?

We have revised our introduction and hope it is now clear that we are not expecting to find a universal constant pattern but want to understand how personality may help explain the pattern (Lines 74-77, 96-101).

L70-71: please explain briefly the choice of this particular aspect of cognition, also in reference to the ongoing debate on whether problem-solving and innovation are indicators of cognitive ability (e.g. Chen et al., 2020; Camacho-Alpízar et al., 2020). While I personally support this view, I believe that in this context the debate should at least be acknowledged.

Thank you, we have added a note about this (Lines 78-79).

L73-74: please explain in what way innovation is crucial to find new resources, it does not transpire clearly from this sentence (“Innovation, the ability to solve novel problems or to find new solutions to an old problems [8], is crucial for locating and using novel resources in a changing environment”) and is not mentioned elsewhere in the intro.

We have added additional explanation of our meaning here (Lines 79-84).

L72-77: these three sentences are a good example of the issues I find with the presentation of this study. The frequent oscillation between biological relevance of the topic addressed here – the link between personality and innovation, which is often mentioned but never addressed in

depth – and methodological aspects. I believe that if the authors find the connection they investigate meaningful in a biological sense, it should be addressed beyond the mention of the one meta-analysis that found that the direction of the relationship varies. And the significance of this relation should be kept separate from the methodological considerations.

Thank you for this suggestion. We have reorganized our introduction to address your concerns (Lines 78-144).

L76-77: please provide support for these statements, as they refer to existing knowledge and previous studies.

We have added the necessary citations to support this statement (Line 87).

L77-83: after describing the variation in results, I suggest making it explicit why this is considered problematic. Why would it be good to have a one-way expectation for the link between personality and problem-solving performance? And do the authors think that given the diversity of morphology, environments, social structure in the listed taxa, such a one-way relationship is really possible?

No, we do not think it would be good (or possible) to have a one-way expectation for the link between personality and problem-solving performance. We have tried to make this more explicit (Lines 96-105).

L89-90: I disagree: personality might affect the performance, the way in which a cognitive task is approached and solved, not the ability itself.

This is a good point. We revised this statement (Line 103-104).

L91-93: And what is the significance of having some personality measures repeatable after an interval of 2 years? What does it mean for zebra finches personality, its evolution, its trade-off with plasticity? And what methodological conclusions can be derived from knowing that zebra finches maintain some of their behavioural responses consistent throughout a significant part of their life?

For zebra finches, two years is a long time span, representing a significant portion of their life, so repeatability of measures after two years suggests traits may be consistent across life stages. If some traits are stable across ontogeny while others are not, the less stable traits may be more plastic and enable zebra finches to respond to fluctuating or challenging environmental conditions. Methodologically, there have been calls for multiple assessments of personality traits on a large time scale (relative to the lifespan of the species) (e.g., Wuerz & Kruger, 2015), so we address this gap.

L101: the fact that a study has not been conducted in a certain species is not a sufficient reason to justify one. Please provide reasons why you consider this species a suitable model to address your question, and why it should provide answers that are valid across taxa, as it seems implied in the introduction and abstract in their current form.

We agree that it is not sufficient alone for a study to have not been conducted in a certain species. We have added additional species justification to the introduction and abstract (Lines 43-45; 53-55; 161-163).

L104-107: "Individuals capable of accessing food resources (such as more dominant individuals), especially when food is ephemeral, may possess an advantage over other individuals to expand the amount or type of food available to them. We filled this gap in knowledge by [...]" I see no gap in knowledge in the previous sentence, and the statement does not really explain the study, especially since foraging or fitness consequences are not assessed.

We have revised this statement (Line 159).

L112-113: Why is it important to have multiple measures of both cognitive performance and personality? What does that add to both fields? A reader should not be forced to Griffin et al. (2015) to start to have answers on this. Please revise the introduction so that the relevance of the study is clear, even to reader who are not familiar with the studies you cite.

We have revised this (and related) statement(s) to avoid “overselling” our multiple measures of cognitive performance (following your comment above), and we have also elaborated on the importance of obtaining multiple measures (Lines 96-105).

Table 1: I am unclear how some of these predictions were developed. If the relationship between personality and innovation varies across species and no previous study was conducted in zebra finches, how did you choose which studies to rely on to inform the predictions (e.g. on dominance?).

This is a great question, especially because some traits we measured here (risk taking and obstinacy) have never been measured in relation to innovation before, to our knowledge. To inform our predictions about the relationship between personality and innovation, we explored much of the literature on personality and innovation, especially in birds. We summarize our findings in Table 1. For dominance, there is evidence for a relationship in both directions, so we did not necessarily predict one or the other.

There is a single prediction clumping together activity, neophobia and aggressiveness; are these traits considered interchangeable? Given the ecological meaning of each, and how very distinct from one another they are, I found it difficult to understand why they are considered together in here – unless this was done in light of the obtained results? If that is the case, please revise, as predictions should arise well before knowing the study outcome.

These traits are not interchangeable, because they represent different axes of personality, but they have been found to be correlated with one another. Other studies (e.g., Sih et al., 2004; Sih & Del Giudice, 2012; Overington et al., 2011; Greenberg, 2003) have found evidence of behavioral syndromes or correlations between activity and/or neophobia and/or aggressiveness, so we originally presented them together for ease of citing these studies, but we have now separated the predictions out by personality trait (Table 1). Also, our own results help confirm that these traits are not interchangeable, nor do they form a behavioral syndrome.

The predictions on risk-taking are mixing the methodological approach (how this trait is measured – in a fraction of studies) with its significance and potential to affect problem-solving performance and innovation propensity.

Unfortunately there have not been any studies looking at the association between risk taking and innovation or obstinacy and innovation (to our knowledge), so we are basing our predictions for these traits on how they are measured in the literature.

Lastly, I have never come across the term “obstinacy” (in the context of studies of personality and cognition). Reading further, I understood it might be akin to what others call “docility”. Like me, maybe other readers are not familiar with this term, so please provide a short definition.

We have added “(latency to catch)” to Table 1.

L135: since there are exceptions to most aspects of the protocols described here, I suggest moving the relevant information to the corresponding section regarding each test and leaving here at the beginning only the descriptions that apply to all test procedures. It would be interesting to know for each test, whether humans were in the testing room or whether the experiment was observed via camera.

We reorganized this section so it is easier to keep track of each test's protocol (Lines 198-273).

L140-142: could you please elaborate a little more on how this protocol helped minimize the isolation stress? Was the bird aware of the cage-mate through sound or smell? I think this aspect is very important and I applaud the authors for thinking of this and implementing it. Explaining how this worked will help future readers to incorporate this concept (if not the exact protocols) in their studies.

We have added detail about the birds being aware of their cage-mate through sound (Lines 230-232). Thank you, we hope this protocol helps inform future personality tests.

L146: based on variation in body mass, metabolism and amount of food ingested prior to the fasting, I argue that food-depriving individuals prior to foraging tasks will have certainly increased motivation, but not standardised it. I suggest replacing "standardised" with "increased" or any similar concept.

This is a great point. We changed "standardize" to "increase" (e.g., Line 230)

L148: a dominance test usually involves some kind of experimental manipulations of dyadic encounters, whereas here it seems to have been an observation of interactions in a multi-bird cage with food-deprived birds. To avoid confusion and misinterpretation of what you actually did (which I think is perfectly fine), I suggest referring to this as "assessment of dominance hierarchies".

We have changed this to "assessment of dominance hierarchies" (Line 212 and throughout manuscript).

Line 153-159: (dominance test) Is latency time to approach the feeder for the first time for each individual not measured?

That is correct. Following Boogert et al. (2006) we did not measure latency to approach the feeder for the first time for each individual; we measured latency to feed.

L163-170: were these parameters also measured in absence of any object? How can you tell that e.g. a longer latency to the feeder is due to the presence of the object otherwise (and not, for example, to differences in general activity)? Also, it might be discussed, whether the novel object was placed close enough to the feeder to prevent the animal from feeding; from the video it looks like it was not adjacent to the feeder but facing it.

Yes, we measured these parameters in the absence of any object (following Boogert et al., 2006). To determine if a significantly different latency to feed is due to the presence of the object and not differences in general activity, we compared mean responses across all 6 days of testing in Trial 1 (no object) to those in Trial 2 (with object), as well as those in Trial 2 (with object) compared to those in Trial 3 (no object) (Boogert et al., 2006). Between each trial there was a 2-minute intertrial interval, and we used approximately the same number of seeds in the feeder in each trial. Wilcoxon rank tests indicated a significant difference between Trial 1 and Trial 2 responses, where Trial 2 had a higher latency to feed ($p < 0.001$), as well as between Trial 2 and Trial 3 responses, where Trial 2 again had a higher latency to feed ($p < 0.001$) (Boogert et al., 2006). The objects were each placed near the feeder but did not prevent the animal from being able to feed (Boogert et al., 2006). We have added this information to the manuscript (Lines 247-248, 397-399).

L185-186 (struggle test): The reader will be wondering, how often the birds are handled in the

daily laboratory routine or for different testing procedures. Are the experimenters known to the animals?

We have added these details (Lines 271-273).

L192-193: please report which is the life expectancy of zebra finches and do not force the reader to search for the cited paper. This information helps the reader put into context your results in terms of long-term repeatability.

We have added this information (Lines 277-278).

L223-230: In the SI, only videos 1-6 are present, so all comments on these tests are based on the written description and figures only.

We apologize. We are not sure why these videos did not play (anyone with the link should have access). Please let us know if you have any difficulty playing them now.

SI Video 1: <https://youtu.be/6Lzifoa9Eyk>

SI Video 2: <https://youtu.be/Pmv7JserD-Y>

SI Video 3: <https://youtu.be/OkfPM30rwPA>

SI Video 4: <https://youtu.be/n6YHfUq3yEQ>

SI Video 5: <https://youtu.be/n9H7qHCoVQg>

SI Video 6: <https://youtu.be/X3LEYBUubQs>

SI Video 7: <https://youtu.be/uTIPmxunu1s>

SI Video 8: <https://youtu.be/0pLh6WNt4b4>

SI Video 9: <https://youtu.be/TKAVXMAMTWQ>

L254: Did the different personality traits correlate with others within and among individuals? Was there a behavioural syndrome that would justify the addition of the cognitive aspects in the collection of behavioural variables recorded so far?

We did not explicitly test for behavioural syndromes within personality traits here, as we were mainly concerned with links between personality traits and innovativeness.

Previous work has laid the foundation for evidence of behavioral syndromes in zebra finch personality (e.g., David et al., 2011), so we sought to expand on this previous work in a new direction.

Also, I could not find clear mentions of whether the performance in the problem-solving tests was repeatable? This is one of the main recommendation by Griffin et al. (2015), which informed many methodological choices in this study. If the cognitive trait is not repeatable, what is the point of investigating its relation to repeatable personality traits?

Repeatability of cognition was not a goal of the current study, but we have now tested repeatability of latency to solve. We found a trend that latency to solve was consistent across tasks ($r = 0.387$, $p = 0.052$). We have added this to the manuscript (Lines 428-429).

Finally, please consider a broader way to statistically analyse correlations for many different tests, a multi-trait analysis or a variable reduction approach.

We used Pearson correlations between measures (i.e., not tests) within the same test in order to determine which measure(s) to retain for further analyses with each trait. We

considered using Principal Components Analysis for this instead of correlations, but given controversy about the validity/subjectivity of interpreting the underlying meaning of principal components (e.g., Budaev, 2010), we decided it would be most conservative and most objective to keep our original correlations.

L265-267: please clarify how these aspects relate to your main questions and hypotheses. Also, did you control for any fixed factors, like for examples sex of the individual or test round or order of presentation? Considering that only a fraction of the animals solved the tasks, not controlling for these aspects puts reliability of results obtained with a relatively low sample size into question.

We have added clarification here (Lines 353, 376-377, 412-413). Please let us know if anything remains unclear. Unfortunately, sex differences were not a focus of the current study, as we wanted to prioritize testing multiple personality traits and their link(s) to cognition, but we have now added tests for sex differences in repeatability of personality traits (Lines 412-420). Also, all animals received the same order of presentation of tests so we could not test for effects of order of presentation (Please see lines 198-199).

Table S2-3: please report all the value for the personality tests that were conducted. Restricting the presentation of the results to these few tests prevents a complete assessment of the results. **If you mean Tables 2-3 (i.e., not supplemental), these are our personality repeatability results and therefore inherently do not include all the values for the personality tests that were conducted, as some of them were not conducted multiple times in 2016 (Table 2) or 2018 (Table 3) (please see our Methods section). Please let us know if you are referring to something else.**

L296-297: do non-solvers also include animals that did not try to solve the tasks? Or did all birds participate?

All individuals approached the apparatuses, but some did not touch the task. Non-solvers includes both of these categories. We have added a note about this (Lines 427-428).

L336-337: my understanding is that the main question of the paper concerns the relationship between some personality traits and problem solving performance. This is again a long discussion of methodological aspects, that does not really address the implications for the observed connection between personality and innovation. Conjecturing on why some of the parameters are not repeatable contributes very little to the discussion of the results in the context of the main questions this study addresses. If I misunderstood which question is prominent for the authors, a revision of the introduction that helps readers focus more clearly on the relevant results will be good.

We have revised our introduction to better explain our two main objectives. Our discussion now matches our introduction, in that we discuss repeatability/methodological aspects and then the connection between personality and innovation.

L386-387: please provide a definition of cognitive style, as this is the first time the term is used and it is not clear why all personality traits should have been involved in all tasks for them to be part of a cognitive style (or perhaps you meant syndrome?)

We have changed this to “cognitive syndrome” (Lines 550).

L398-399: please discuss why you think this was the case. Stating that personality traits were

weakly connected to problem solving in a task-dependent way is a repetition of the results and does not add to the current understanding of the relationship.

Thank you for this suggestion. We have elaborated on our discussion point (Lines 575-576).

L411-412: The conclusion of the study is weak and not comprehensive to the reader. A conclusion might include why it is important to consider your work and multiple personality traits for future studies, and what is the biological meaning of the connection.

We believe we have contributed a long-term assessment using a comprehensive suite of traits, but we agree that at times it makes for a non-straightforward discussion of our results, since our results are specific to each trait and task. We have added additional discussion about the importance of our work (Lines 593-600).

Minor comments

Table 1: I suggest an extra column filled with + or – signs (or alternative signs) that indicate e.g. your expecting increasing aggressiveness to predict increasing problem-solving ability.

This is a great suggestion, thank you. We added a column of directional signs (Table 1).

L136: “We conducted one behavioural assessment per day” per animal I assume?

Yes, thank you. We have added this detail (Line 198).

L173 (mirror test) “a small piece of millet” can be misunderstood as small grain, better might be: piece of foxtail millet or just millet grains or millet.

We replaced this with “millet” (Lines 251-252, 505).

L340-341: something seems to be wrong in this sentence “A post-hoc nonparametric Wilcoxon test indicated a significantly lower time to catch on average in 2018 ($\bar{x} = 21.48$ s) compared to 2016 ($\bar{x} = 14.73$ s)” – shouldn’t 14s be a lower time to catch than 21s?

Thank you for bringing this to our attention. We accidentally swapped the values, and have now corrected this (Lines 487-488).

L358-359: habituation to captivity? Weren’t they born in captivity? Probably adjustment to new housing conditions is what is intended?

Yes, we have rephrased this (Line 512).

SI Document 1: those exact measures might be reported directly in the methods section.

We have moved them to our Fig. 2 caption (following your and Reviewer 1’s comment).

Appendix C

Dear Dr. Mathot,

We are grateful for the opportunity to resubmit our substantially revised manuscript (RSOS-212001). We also appreciate the extension on our revisions. Please find below a point-by-point response (in bold) to the comments provided by you and the reviewers, with line number references to our *Track Changes PDF* document. In nearly every case, we have implemented the suggested changes, and we feel the paper has significantly improved as a result. We hope that you agree.

Thank you for considering our revised manuscript for publication.

Sincerely,

Lisa P. Barrett

Corresponding Author, on behalf of all co-authors

Dear Dr Barrett

The Editors assigned to your paper RSOS-212001 "Links between personality traits and problem-solving performance in zebra finches (*Taeniopygia guttata*)" have now received comments from reviewers and would like you to revise the paper in accordance with the reviewer comments and any comments from the Editors. Please note this decision does not guarantee eventual acceptance.

Please submit your revised manuscript and required files (see below) no later than 21 days from today's (ie 02-Feb-2022) date. Note: the ScholarOne system will 'lock' if submission of the revision is attempted 21 or more days after the deadline. If you do not think you will be able to meet this deadline please contact the editorial office immediately.

on behalf of Dr Kimberley Mathot (Associate Editor) and Kevin Padian (Subject Editor)
openscience@royalsociety.org

Editor comments:

Thank you for your resubmission, and for your diligence in addressing the comments of the reviewers. As you see from the AE's comments there are still some issues to resolve that I hope you can take care of easily. If you need more time than the allotted window, please contact the editorial office. I remind you that your next version needs to satisfy all concerns of the reviewers and AE, or we will be unable to consider it further. Thanks and best wishes.

Associate Editor Comments to Author (Dr Kimberley Mathot):

Associate Editor

Comments to the Author:

Thank you for submitting a revised version of your manuscript for consideration at RSOS. The revised manuscript has now been reviewed by two of the referees who commented on the original submission. Both referees feel that the current version of the manuscript is much improved. Specifically, the aims of the study are laid out more clearly in the introduction, and the methods are described in better detail. Both referees still have a number of points that I would like to see addressed before the manuscript can be accepted. Most of these are minor.

We are glad the referees feel the current version is much improved. We have now addressed all of their points, further improving our manuscript.

However, one of the more significant points I would like to see addressed is the concern raised about whether (or the extent to which) the three cognitive tasks are independent measures of cognitive ability. There are various ways that you could address this. In fact, part of your results seem to address this, but it's not entirely clear to me the models from which these results were derived (lines 427-429). I would suggest a two-phase approach to analyze the non-independence of the three cognitive tasks.

1) is the probability of solving repeatable

$\text{glmer}(\text{solved} \sim \text{task_type} + (1|ID), \text{family} = \text{binomial})$, where solved is (1 = yes, 0 = no), and task type is "string pull", "wire removal" or "lid flip".

From this model you can assess the repeatability of solving yes/no as a way of inferring the extent to which the tasks may be independent.

2) for birds that solved, is the time to solve repeatable.

$\text{glmer}(\text{time to solve} \sim \text{task type} + (1|ID), \text{family} = \text{poisson})$.

Note, you may assign max time values for non-solving birds. You may also find that transforming the data and then modelling gaussian errors works better if the poisson models are

overdispersed.

Note that any degree of repeatability will give you information about the potential non-independent between these different test types. Also- it is important that other predictors are not included in these models (e.g., behaviours, sex), as that will erode part of the among-individual variation. Regardless of the finding, it would be worth explicitly acknowledging the degree of (non-)independence in the results and discussion.

Thank you for clarifying about this point. We did analyze these models (without other predictors) in our previous revision and found that the responses of 1) binomial solve yes/no and 2) time to solve (after checking for overdispersion) were not significantly repeatable. We found a strong trend in time (latency) to solve, however. We apologize that this was not clear. We have now clarified this in our methods (Lines 372-374) and results (Lines 418-420). We also added a note about these results to our discussion section (Lines 532-539). Please let us know if any additional information would be helpful.

Reviewer #1 also points out that the analytical decision to use means for behavioural traits rather than the repeated measures comes with several limitations. They are absolutely correct. However, given your data structure, the analytical approach that would allow you to use repeated measures of behaviour with a single measure per cognitive task would require a multivariate model, and your sampling design (number of individuals, number of repeats per individual) is not sufficient for you to have the power to do this. Instead, I would ask that you address this point by acknowledging the consequences that using a single measure per individual can have for parameter estimates. Referee #1 has provided you with some relevant references to address this.

Thank you for this helpful suggestion. We understand that using means for behavioural traits instead of repeated measures comes with limitations. We have added an acknowledgement about the consequences of using a single measure per individual (discussed in Brommer (2013)) to our discussion (Lines 550-553).

Brommer, J. E. (2013). On between-individual and residual (co) variances in the study of animal personality: are you willing to take the “individual gambit”? *Behavioral Ecology and Sociobiology*, 67(6), 1027-1032.

In addition to the comments provided by the referees, I have the following comments I would like you to address in the revised version:

1. Please refrain from referring to specific behavioural traits as “personality traits”. Personality is a statistical property of a population (i.e., repeatable among-individual variation) – and any behaviour for which such repeatable among-individual variation exists can be studied in an animal personality framework. Further – studying a trait that is commonly studied in animal personality (e.g., boldness, neophobia) cannot be assumed a priori to harbour significant among-individual variation. You assess the repeatability of the traits you studied – so it won’t change any of your conclusions, but the terminology should be corrected.

For example, on line 166 you write, “are personality traits repeatable not only over the short-term but also over the relatively longer-term?”. By definition, the answer must be yes otherwise

the traits cannot be studied in an animal personality framework. Instead, you can write “1) we estimated both the short- and long-term repeatability of X, Y, Z, and asked whether repeatable variation in X, Y, Z as associated with differences in problem-solving performance.

Thank you for bringing up this point. We appreciate your comment, and we have corrected our terminology here (Lines 146-150) and throughout the manuscript (e.g., 47, 114-115, 145, 160, 450-451, 509, 510, 557).

2. For Poisson models (Line 381)– please confirm that you checked for overdispersion.

We have now checked for overdispersion and added the necessary detail to our Methods (Line 341-342, 364) and updated our results (Table 3, Table 4).

Reviewer comments to Author:

Reviewer: 2

Comments to the Author(s)

In general, the authors carefully attended to my previous comments and the current version of the manuscript has improved a lot. Especially the introduction now makes the study goals clear and the methods are much clearer now. I am left with one major concern and a handful of smaller suggestions to improve readability.

Thank you very much for taking the time to review our substantially revised manuscript. We are glad the goals and methods are much clearer now. We hope you agree we have improved readability in our newest version.

All lines refer to the manuscript with highlighted changes and to the original line numbers included in the word or similar program.

My major concern is the analysis of the cognitive tests and their association with personality traits. At the moment, the reader is left to wonder whether the three cognitive tests really all reflect problem-solving behaviour or describe various different cognitive traits in zebra finches. I suggest to investigate this in more detail (see my comment to line 422). This knowledge should then inform how associations with the different personality measures are analysed. At the moment the probability to solve is analysed separately for all three tests while the average latency to solve across all three tests is considered in 1 model. If all three tests indeed reflect problem-solving behaviour, then one model including all three tests should investigate personality associations for solving probability and a second model should investigate personality and solving latency.

Thank you for this feedback. We agree that it is important to investigate whether the three cognitive tasks are independent assessments of cognitive ability. Please see our above response to the editor regarding this concern and how we have addressed it. To summarize, we do not have evidence of repeatability of our binomial problem-solving success variable or our problem-solving latency variable across cognitive tasks. Therefore, it does appear that the tasks are independent assessments of cognition.

However, we cannot definitively state the exact cognitive abilities that each task assessed without further study.

In addition, I strongly suggest to use a mixed model framework then instead of averaging across several test situations. Otherwise, it will not be possible to separate within- from between individual effects (see for a similar discussion on animal personality and behavioural syndromes for example Brommer (2013)). If, however, the three tests measure different aspects of cognition, the analyses should be split up for the tests consistently, i.e. the model explaining average latency should not be used. Based on these analyses, I would then also advice to re-write the second part of the discussion starting in line 527. At the moment, this part of the introduction still leaves the reader wondering what we have learnt about the association of personality and cognition/ problem-solving in particular.

Brommer, J. E. (2013). On between-individual and residual (co) variances in the study of animal personality: are you willing to take the “individual gambit”?. Behavioral Ecology and Sociobiology, 67(6), 1027-1032.

Thank you for this suggestion. Unfortunately we are limited by our data structure, but we understand that using means for behavioral traits instead of repeated measures comes with limitations. We do specifically address this limitation in our revised discussion (Lines 550-553). Please see the Editor’s relevant comment and our response about this.

- Line 36-37: this sentence does not make sense here since cognitive abilities have not yet been introduced. Move to line 41

We have moved this sentence to later, after cognitive abilities have been introduced (Lines 39-40, 42-43), and we have revised the abstract to introduce cognition earlier (Lines 37-39). We also substantially cut text to fit within the word limit.

- Line 41: measuring instead of measured

We made this change (Line 46).

- Check consistency of referencing, e.g. line 84, change in style

We apologize for this error. We checked consistency of referencing and made the required changes (Lines 84, 156).

- Line 113: add “,” after: Thus

We added a comma (Line 121).

- Line 147-148: personality does not fall under animal cognition, please rephrase sentence

We rephrased this sentence to make a distinction between cognition studies and personality studies (Lines 126-127).

- Line 193-194: I cannot judge if permits would needed to have been acquired under the local laws. In the European Union, experiments such as performed here would need a formal application to the responsible authorities

Our university IACUC did not recommend any other formal applications and we are not aware of any required for this type of work.

- Line 246-247: is there something missing from this sentence? I don't understand the meaning. In addition, the sentence comparing response between trials is copy-pasted in the results section. I think it makes more sense there and should be deleted here.

We agree. We removed this sentence here (Line 218) since it belongs later where we originally included it (Lines 384-387).

- Line 353: there is a verb missing in this sentence

Thank you for pointing this out. We have inserted the verb ("calculated") (Line 337).

- Line 368-369: what kind of correlation was used here? And how were repeated measures from individuals dealt with?

We have added that a Pearson correlation of average responses across trials were used (Line 351-352).

- Line 399: why is there a citation in the results section? I think the results reported in this sentence were collected in this study?

We removed citations in the results section (Line 387, 406).

In addition, I do not understand why a separate Wilcoxon test was performed here. The result whether or not trial performance differed from each other should be available in the model estimating the repeatability since "trial" was included as fixed effect. Or was "day 1-6" included in the model here instead of trial (compare description lines 242-243)? In addition, I'm wondering, to estimate the repeatability here, were both daily trials (i.e. without object and with object) included in the analysis? This might explain a non-significant repeatability since these are quite different situations.

The model estimating repeatability with "Trial" as a fixed effect was indeed "Day 1-6" (we use "Trial" to simplify, but perhaps this was more confusing). We have corrected this to read "Day 1-6" (Lines 343, 344-345). To answer your second question: No, trials with and without objects were not both included in the analysis. Only trials with the object were included for repeatability models. We have added this detail (Lines 346-347).

- Lines 412-420: I suggest to delete this section since just comparing confidence intervals like this is not sufficient to determine if the sexes differ in repeatability or not. As the investigation of potential sex-differences was not a main aim of the study, losing this paragraph will not affect the general conclusions.

We included this additional analysis at the request of reviewers during our first revision. Please see Jolles et al. (2019) for similar methods. However, we are happy to remove this section if the Editor would like us to.

Jolles, J.W., Briggs, H.D., Araya-Ajoy, Y.G., & Boogert, N.J. (2019). Personality, plasticity and predictability in sticklebacks: bold fish are less plastic and more predictable than shy fish. *Animal Behaviour*, 154, 193-202.

- Line 422: I suggest to additionally test for repeatability after excluding all individuals that never approached the test setup. Currently, non-solvers could include animals that lack motivation to interact with the test setup as well as animals who are really just bad innovators and therefore score the maximum latency. In addition, I suggest to run a model checking the probability to solve, i.e. assuming a binomial distribution to inform us more about the generality of the tests used here to investigate problem-solving behaviour.

All individuals approached the apparatuses, so we cannot exclude individuals that never approached the test setup. We have included all individuals at the request of the Editor (please see the Editor's suggested analyses about repeatability of problem-solving performance above). In our previous revision we ran a model checking the probability to solve (using a binomial distribution) and found that whether or not a bird solved a task was not repeatable across tasks. We have now clarified this in our methods (Lines 372-377) and results (Lines 418-420). We also added a note about these results to our discussion section (Lines 532-539).

- Line 430: please test for variance inflation factors to exclude the possibility that covariation between personality traits of different tests influence the model outcome.

We have tested for variance inflation factors to exclude this possibility. We added this detail to our methods (Lines 369-371).

- Line 522-524: I suggest to delete this sentence since the corresponding statistical analysis is not appropriate to test sex-differences in repeatability (see also my comment to lines 412-420).

Please see our comment above. We are happy to remove this if the Editor would like us to.

Reviewer: 3

Comments to the Author(s)

In RSOS-212001 the authors revised their previous manuscript describing a study on long-term repeatability of personality traits, and their link to innovative problem solving. The revisions to the text have, in my opinion, improved the readability of the text, clarified the aims and research questions, and provided an overall better structure for the manuscript. Comments from the previous round of review have been attended, in terms of additional information and clarifications. Methods are now well described and easy to follow.

However, I am afraid that when it comes to the most substantial points raised in the previous round of review, the current revision is not complete or appropriate, making it very difficult to re-evaluate the merit of this contribution. I think that the alternative analytical approaches suggested would have indeed implied an advancement in the way these type of data are used to answer timely questions. The approach the authors chose might have been used in the past, but it does not mean it is the most suited for such a complex dataset as this.

We are very grateful that you have reviewed our revised manuscript, and we are glad you think the revisions have improved readability and structure. We understand that you still have concerns about our analytical approaches. Unfortunately, there were some approaches that were previously suggested by the reviewers that we could not implement due to the particularities of our data. For example, we considered Reviewer 2's suggestion from the previous revision to use a multivariate mixed-effect modelling approach. Unfortunately, it was not appropriate for our data for a number of reasons listed in our previous response to reviewers. To help explain our reasoning for our analytical approach, we revised the language in our previous revision to clarify that we are not assuming causal relationships between variables. We have also tried to address any remaining concerns in this revision.

Additionally, my concern about the multiplicity of the cognitive tasks was not addressed in a convincing manner, in my opinion. I did (and still do) think that the three cognitive tasks used here represent variations of the same test, a removal of something to access the food reward visible through a transparent barrier. The three tasks were targeting the same cognitive ability, and relied on similar cognitive mechanisms, as the authors stated in the first draft. The variation was in the non-cognitive aspects such as motor actions required to solve them. Now the authors state that slightly different cognitive aspects might have been involved in the three tasks, and that is why they were presented. They list the fact that sometimes dominance and sometimes aggressiveness were weakly correlated with success, in support of different cognitive aspects being required in each task (L574-577). I consider this approach quite problematic. First of all, if it is not known which aspects of innovative problem solving the three tasks address, this aspect should likely be illuminated before using such tasks to investigate the connection between problem solving and non-cognitive aspects like personality. Secondly, I do not see how non-cognitive aspects (be it motor diversity or personality) involved in the performance might constitute support for the three tasks being different from a cognitive perspective. In sum, I still think calling these variants "multiple cognitive tasks" misleading, both in describing the width of the study and especially in the results interpretation and recommendation for the future. That it is a repeated test, like the behavioural ones, is agreed. But that these are diverse, multiple cognitive tasks addressing innovation in different ways, is not.

Thank you for your thoughts on this important topic. We agree that we could have made our description of the three tasks and their relatedness more clear. As the editor suggested, we did assess whether performance on the three tasks was repeatable across individuals, specifically to determine how similar these tasks are. Our analyses of repeatability of success on the tasks suggest that performance was not repeatable across tasks. Therefore, we do think the three tasks are different in that individuals who excelled in one task did not necessarily solve the others. We have clarified this finding in our discussion section (Lines 532-539).

Despite individual performance not being repeatable across the three tasks, we cannot definitely state that these tasks differ from one another with regard to the specific cognitive abilities they are testing. Therefore, we have replaced “multiple cognitive tasks” to “repeated tests” (Line 575-576). We also changed “multiple problem-solving tasks” and “three novel foraging problem-solving tasks” to “repeated problem-solving tests” (e.g., Line 45-46, 53 etc.). We also added a caveat to our discussion when interpreting our results: we now note that more work is needed to determine if the tasks differ from one another from a cognitive perspective (Lines 538-539).

Finally, while the general readability has increased, the question of why and how personality should be linked to cognitive performance is still dealt with quite superficially, and does not go beyond the citation of previous studies stating that such links exist and are to be expected.

We are glad that the general readability has increased, and we apologize for the lack of elaboration on why and how personality should be linked to cognitive performance. We have now added an explicit explanation of how personality may correlate with innovation in order to better illustrate in detail which aspects of personality might covary with cognitive performance and through which mechanisms (Lines 87-93, 96-100, 102-104). We also explain this link in our original predictions (Lines 152-159).

Specific comments

L85-87: There is still no explicit explanation of how personality may correlate with innovation, except for mentions of the review. I think it is important to illustrate in detail which aspects of personality might co-vary with cognitive performance, and through which mechanisms. Listing examples (e.g. L91-97) is not quite the same as providing an explanation for the patterns.

Please see our response above. We hope you agree that we have provided an explanation for the patterns, but please let us know if it would be helpful to provide any other information.

L107-109: This clarifies the study aims better. But there is still no biological reason for wanting to know more about long-term repeatability of behavioural responses.

We are glad this clarifies the study aims better. We have also elaborated on our biological reasoning for our aims of studying long-term repeatability of behavioral responses (Lines 118-119, 123).

L151-155: In the previous sentence the authors describe how dominant individuals that arrive at the food first, have more potential to innovate. But the "necessity drives innovation" hypothesis describes exactly the opposite scenario, with less competitive animals having to resort to innovation to compensate for restricted access to common food sources. In the rest of the text the necessity-drives-innovation hypothesis is correctly referred to, so it might have been a temporary mix-up here.

Here we attempt to explain that the relationship between dominance and innovation could go either way: more dominant individuals, or alternatively, *less* dominant individuals (via necessity drives innovation hypothesis), could be better problem solvers. We have clarified this section (Lines 134-137).

Additionally, dominance is not a synonym of aggressiveness in many species - but could be that in zebra finches more aggressive animals (and not e.g. older or bigger) are at the top of the hierarchy. If this is the case here, it should be stated, or the logical progression of arguments is hard to follow.

This is a great point. Yes, previous work in zebra finches has shown that more dominant birds are thought to be more aggressive and gain access to a single feeder (and not bigger/in better body condition) (David et al., 2011). We have clarified this (Lines 132-134).

David, M., Auclair, Y., & Cezilly, F. (2011). Personality predicts social dominance in female zebra finches, *Taeniopygia guttata*, in a feeding context. *Animal Behaviour*, 81, 219-224.

L266-270: Mentions of the ecological relevance if this test appear the very last time this is mentioned in the discussion (L563-566). Until then it is not clear at all what is the supposed ecological relevance of the behaviours measured in this test. I am not inferring that there is none, just pointing out that this information is missing and it gets in the way of understanding the significance of the results.

We moved this mention of ecological relevance of this test to our Methods (Lines 240-242, 554-556).

Further, in my previous round of comments I asked about the existence of studies validating these tests in an ecological context. In order to talk about coevolution with cognition and selective pressure on certain personality traits, we need to know whether the behaviours measured in the lab indeed translate into real-world responses to environmental stimuli. Do these tests predict how zebra finches respond to novelty, startling events, etc. in the wild?

We agree that it is important to know whether the behaviors measured in the lab translate into real-world responses to environmental stimuli. Unfortunately there are not many studies validating these tests in zebra finches in an ecological context. One relevant study compared exploratory behavior in a captive, novel environment to exploration of feeders in the wild and found they were not related; this study also found that neither exploratory tendency nor sociality predicted reproductive success (McCowan et al., 2015). We are not aware of any other studies of wild zebra finch personality, but there have been some studies validating these tests in other species (e.g., great tits) in the wild. In our first response to reviewers, we noted that we added the following relevant caveat to our discussion: "We also note that although we carefully chose assessments used in the animal personality literature [20, 23-25, 37, 38], not all of them have been validated in an ecological setting, and so it is possible we were not measuring the traits we intended to measure," following Carter et al. (2013). Additionally, in this second revision, we have added more information about whether/how these tests are thought to predict how zebra finches respond in the wild into our methods (Lines 183-184, 198-200, 206-207, 222-223, 233-234), but unfortunately many tests have not yet been validated in the wild. For example, obstinacy has not been experimentally tested in terms of its effects on survival in zebra finches, but it has been tested in captive zebra finches because it is thought to relate to escape behavior (David et al., 2011). Therefore we cannot say for sure whether these tests predict how zebra finches respond in the wild, and so we also added a recommendation for future work to validate these tests (Lines

501-503). Please let us know if you think there is any additional information we need to provide.

Carter, A.J., Feeney, W.E., Marshall, H.H., Cowlshaw, G., & Heinsohn, R. (2013). Animal personality: what are behavioural ecologists measuring? *Biol. Rev.*, 89, 465-576.

David, M., Auclair, Y., & Cezilly, F. (2011). Personality predicts social dominance in female zebra finches, *Taeniopygia guttata*, in a feeding context. *Animal Behaviour*, 81, 219-224.

McCowan, L.S.C., Mainwaring, M.C., Prior, N.H., & Griffith, S.C. (2015). Personality in the wild zebra finch: exploration, sociality, and reproduction. *Behavioral Ecology*, 26, 3, 735-746.

L412: the other part of Rew. 1 suggestion, concerning links between personality and cognitive abilities being sometimes sex-specific, was not addressed either in the ms or in the response.

We apologize for the confusion. In our previous revision, we tested for sex differences in repeatability of personality at the request of reviewers. In the current revision, the first reviewer ("Reviewer 2") recommends we delete the section on sex differences (which we are happy to do if the Editor would like us to (please see above: "...the investigation of potential sex-differences was not a main aim of the study, losing this paragraph will not affect the general conclusions")). We have now added interaction terms of sex and each personality trait into our GLMs relating personality to problem-solving performance, and we did not find any sex differences (please see Lines 374-377, 433-435).

Tables 3-4: if I understood correctly, repeatabilities reported here are adjusted for Trial and (where possible) Year. So it would be good to state that these values are for adjusted repeatabilities.

Yes, thank you. We have added this detail (Lines 390, 397).

L478-480: One problem the field of animal personality is facing, is indeed the lack of consistent terminology. What here (and indeed in a few other papers) is called obstinacy, i.e. the response to human capture or handling, is more often described as "docility" and has received more attention than here implied (e.g. Dingemanse and Réale, 2005; Petelle et al., 2013, 2015; Hall et al, 2015; Guenther et al., 2018).

This is a great point that merits further discussion. We have clarified that obstinacy is commonly referred to as docility, and we added the references you recommended (Line 240, 546-547). We note, however, that none of these studies investigated the link between docility/obstinacy and innovation (or over such a long time period (Line 461-462)).

L489-491: I do not think this is what the results show. Most variables of each test were repeatable.

We have clarified that here we are referring to our results pertaining to the two variables of the Struggle Test (Line 473).

L563-566: This arrives quite late (see my comment above). Additionally, stating that only another study on something exists is quite risky (e.g. see Edwards et al. 2017, for a study on repeatability and heritability of obstinacy in warblers).

Thank you for bringing this to our attention. We rephrased our statement about previous studies of obstinacy (Lines 549-550), and we moved our statement about the ecological relevance of obstinacy to earlier in the manuscript (Lines 240-242, 554-556).

L639 (very minor): I believe a comment escaped a final edit of the text.

We apologize for this error. We removed this track change comment.